# Tetrapod sperm length evolution in relation to body mass is shaped by multiple trade-offs

Loren Koçillari [1,2,3,7], Silvia Cattelan [4,5,7] ✉, Maria Berica Rasotto[4], Flavio Seno[2], Amos Maritan [2,6] & Andrea Pilastro [4,6]

Sperm length is highly variable across species and many questions about its variation remain open. Although variation in body mass may affect sperm length evolution through its influence on multiple factors, the extent to which sperm length variation is linked to body mass remains elusive. Here, we use the Pareto multi-task evolution framework to investigate the relationship between sperm length and body mass across tetrapods. We find that tetrapods occupy a triangular Pareto front, indicating that trade-offs shape the evolution of sperm length in relation to body mass. By exploring the factors predicted to influence sperm length evolution, we find that sperm length evolution is mainly driven by sperm competition and clutch size, rather than by genome size. Moreover, the triangular Pareto front is maintained within endotherms, internal fertilizers, mammals and birds, suggesting similar evolutionary trade-offs within tetrapods. Finally, we demonstrate that the Pareto front is robust to phylogenetic dependencies and finite sampling bias. Our findings provide insights into the evolutionary mechanisms driving interspecific sperm length variation and highlight the importance of considering multiple trade-offs in optimizing reproductive traits.

*"[…] How does sperm form relate to function? How do different fertilisation environments shape sperm phenotypes, such as fresh- versus salt-water, or hot- versus cold blooded reproductive tracts? How does sperm morphology co-vary with body size, or genome size?"* Matt Gage – Nature Ecology and Evolution 5, pages 1064–1065 (2021)

In a recent commentary on the most comprehensive analysis of sperm size variation across the animal kingdom published to date[1], Matt Gage[2] concluded that despite intensive research shedding light on the evolution of the large variability in sperm size and form, many questions remain open. Sperm size, and in particular sperm length, the most common measure used for sperm size (to which we will refer

hereafter), varies enormously between taxa[1], ranging from less than 10 μm (in some wasps) to nearly $10^5$ μm (in some fruit flies)[3,4]. The most extensively studied taxa are arthropods and vertebrates, particularly tetrapods. The shortest and longest sperm recorded so far are both found in arthropods, while sperm length in vertebrates does not show the same level of variation, ranging from slightly more than 10 μm (in external fertilizers such as bony fishes and frogs) to approximately $10^3$ μm (in some urodeles)[4]. It has been recently demonstrated on a large taxonomical scale that the fertilization mode represents an important source of variation, with external fertilizers having, on average, shorter sperm than internal fertilizers[1,5]. Moreover, sperm competition (i.e., when sperm from different males compete to fertilize the same eggs)[6–8] leads to the evolution of (1) the production of

¹Istituto Italiano di Tecnologia, 38068 Rovereto, Italy. ²Department of Physics and Astronomy, Section INFN, University of Padova, 35131 Padova, Italy. ³Institute for Neural Information Processing, Center for Molecular Neurobiology Hamburg (ZMNH), University Medical Center Hamburg-Eppendorf (UKE), D-20251 Hamburg, Germany. ⁴Department of Biology, University of Padova, 35121 Padova, Italy. ⁵Fritz Lipmann Institute–Leibniz Institute on Aging, 07745 Jena, Germany. ⁶National Biodiversity Future Center, 90133 Palermo, Italy. ⁷These authors contributed equally: Loren Koçillari, Silvia Cattelan. ✉ e-mail: silvia.cattelan@leibniz-fli.de

more sperm, via increasing testes mass, and (2) longer sperm[9] (with exceptions limited to a few taxa reviewed in ref. [10]). The evolution of larger testes in response to sperm competition has received such a broad empirical support that relative testes mass is commonly used as a proxy for sperm competition[9]. In contrast, positive selection for longer sperm has been just recently recognized as a general trend[9], probably because the mechanisms leading to the evolution of longer sperm differ between taxa. In small animals, sperm actively interact in the female reproductive tract and longer sperm have a direct competitive advantage by displacing rival male sperm or saturating female storage organs[6,11]. Simultaneously, the risk for sperm dilution, which is predicted to favor the evolution of sperm number at the expense of sperm length, is relatively less important in small animals compared to large animals[6]. On the other side, in large animals, sperm have to move via a longer female reproductive tract[12], which may favor the evolution of longer sperm via selection for faster sperm swimming speed[13]. As a result, interspecific variation in body size seems crucial in mediating different processes for the evolution of sperm length under equivalent levels of sperm competition[14].

Variation in body size may also affect sperm length evolution through its influence on other factors which, in turn, may be expected to influence sperm length evolution. Body size explains a great proportion of the variation in life-history strategies observed across animals[15], which result from the interplay between body mass and population density[16], longevity[17] and reproductive strategies[18,19], including fertilization mode and sperm competition[20,21]. For instance, clutch/litter size, which covaries with body size in most vertebrate classes (albeit in different directions[22–24]), may influence the overall investment in sperm number and hence the ratio of sperm number/length[24,25]. In frogs, sperm length covaries with egg size[26,27], which in turn is influenced positively by body size and negatively by clutch size[28]. In contrast, clutch size in birds is positively correlated with sperm storage duration[29,30], which in turn correlates with the size of female storage organs[29]. Since the length of female storage organs is positively correlated with sperm length[30,31], probably because longer sperm are more efficient in persisting within the female storage organs, an overall positive association between sperm length and clutch size is expected. However, how and whether body size and clutch size interact affecting the evolution of sperm length at a large taxonomical scale is still unknown, probably due to the complex and contrasting influence that body size exerts on the association between clutch size and egg size in external[28] and internal fertilizers[32].

Body size may affect the evolution of sperm length through its link with metabolic rate and genome size. Large species have a lower metabolic rate compared to small-sized species[33,34], which has been suggested to influence the production of long sperm due to a low cellular activity[35,36]. This hypothesis would help explain why small mammals, which have high metabolic rate, evolve long sperm in response to high levels of sperm competition, while the same relationship is not observed in large-bodied mammals, which have a relatively lower metabolic rate[35,37]. Body size has also been shown to be positively associated with genome size, at least in some groups, e.g., ref. [38]. Genome size, in turn, is strongly correlated with cellular size from yeast[39] to multicellular organisms[40–42]. The relationship between genome size and cell size also applies to gametes (both eggs and sperm)[43] (but see ref. [44]). Sperm cells with a large genome may require a long flagellum to compensate for the drag caused by the bigger sperm head, especially in large species in which sperm have to move via a longer female reproductive tract. Sperm length in relation to body size may thus result from the contrasting effects of metabolic rate and genome size[45]. In large animals, low metabolic rate may constrain the evolution of sperm length while a large genome size may favor the evolution of long sperm. How these two contrasting hypotheses have shaped the evolution of sperm length in relation to body size in animals is unknown. In conclusion, despite the increasing interest in the

evolution of sperm length and the several direct and indirect mechanisms through which body size may be potentially linked to sperm length through complex trade-offs, how and to which extent sperm length variation is linked to body mass remains relatively poorly known[2].

Here, we hypothesized that the relationship between body size and sperm length in tetrapods results from trade-offs among multiple factors with potential contrasting effects. We tested our hypothesis in the context of Pareto Optimality. The Pareto optimization approach has been used across various biological systems, including gene expression[46], tumors[47], proteins[48,49], trait morphology[50–54], neural systems[55,56] and behavioral traits[57,58]. Pareto Optimality predicts that in the presence of evolutionary trade-offs across multiple tasks, phenotypes are distributed along low-dimensional geometrical patterns, or Pareto fronts, in the trait space[59]. The geometric shapes of these Pareto fronts depend on the number of tasks in trade-off (see Fig. 1). In the case of two tasks, the Pareto front becomes a line segment; for three tasks, it forms a triangle; and for more tasks, the Pareto front becomes a convex polygon, whose number of vertices corresponds to the number of tasks. Therefore, if the distribution of phenotypes fall within a polygon, the number of vertices is expected to reflect the number of trade-offs (Fig. 1). If phenotypes are uniformly distributed as irregular clouds of points in the trait space, no trade-offs are expected. Trade-offs can then be inferred by investigating other phenotypic traits that are enriched at the vertices of the polygons– the so-called archetypes (Fig. 1).

In this work, we used the Pareto Task Inference (ParTI)[47,50] to analyze an extensive dataset of 1388 tetrapods, which included 231 amphibians, 115 reptiles, 399 birds, and 643 mammals. We first tested whether tetrapods lie on a Pareto front in the trait space of sperm length and body mass. We show that tetrapods are distributed within a triangular Pareto front, as expected if trade-offs constrain the evolution of sperm length in relation to body mass. Secondly, we tested whether the species populating the three vertices of the Pareto front differed in clutch/litter size, sperm competition, and genome size by conducting a feature enrichment analysis. We find that sperm length evolution is mainly driven by sperm competition and clutch size, rather than by genome size. Third, we show that the sperm length–body mass Pareto front is maintained within functional groups, based on thermoregulation mode (endothermic/ectothermic tetrapods), fertilization mode (internal/external fertilizer species), and across taxonomic groups (the four classes of tetrapods). Finally, we demonstrate that the Pareto front is robust to phylogenetic dependencies and sampling bias.

## Results
### Tetrapods fall within a triangular Pareto front in the trait space of body mass and sperm length
We analyzed an extensive dataset of 1388 tetrapods that we compiled from publicly available sources (see "Methods"). For each species we collected data on several continuous features. The features include sperm length (μm)[60], body mass (g), clutch/litter size, relative testes mass as a proxy for sperm competition[9], and genome size (see Methods). To test whether the sperm length-body mass relationship in tetrapods results from trade-offs between multiple tasks, we searched for the presence of Pareto fronts in the space of body mass (BM) and sperm length (SL). The distribution of data in the double logarithmic space of BM and SL exhibited a triangular geometric shape (Fig. 2a). Therefore, we hypothesized that a triangular Pareto front would best fit the data. Using the $t$-ratio test (see "Methods"), we found a significant triangular Pareto front ($t$-ratio test, $p < 0.001$, Fig. 2a and Supplementary Fig. 2). We then assessed the goodness of fit of the triangular Pareto front to the data compared to other possible Pareto fronts with different geometric shapes (line segments, squares, etc.). We used the Principal Convex Hull/Archetypal analysis (PCHA)[61] to fit the distribution of data with the Pareto fronts with $n = 2,...,5$ number of vertices and found that the Pareto front with three vertices minimizes

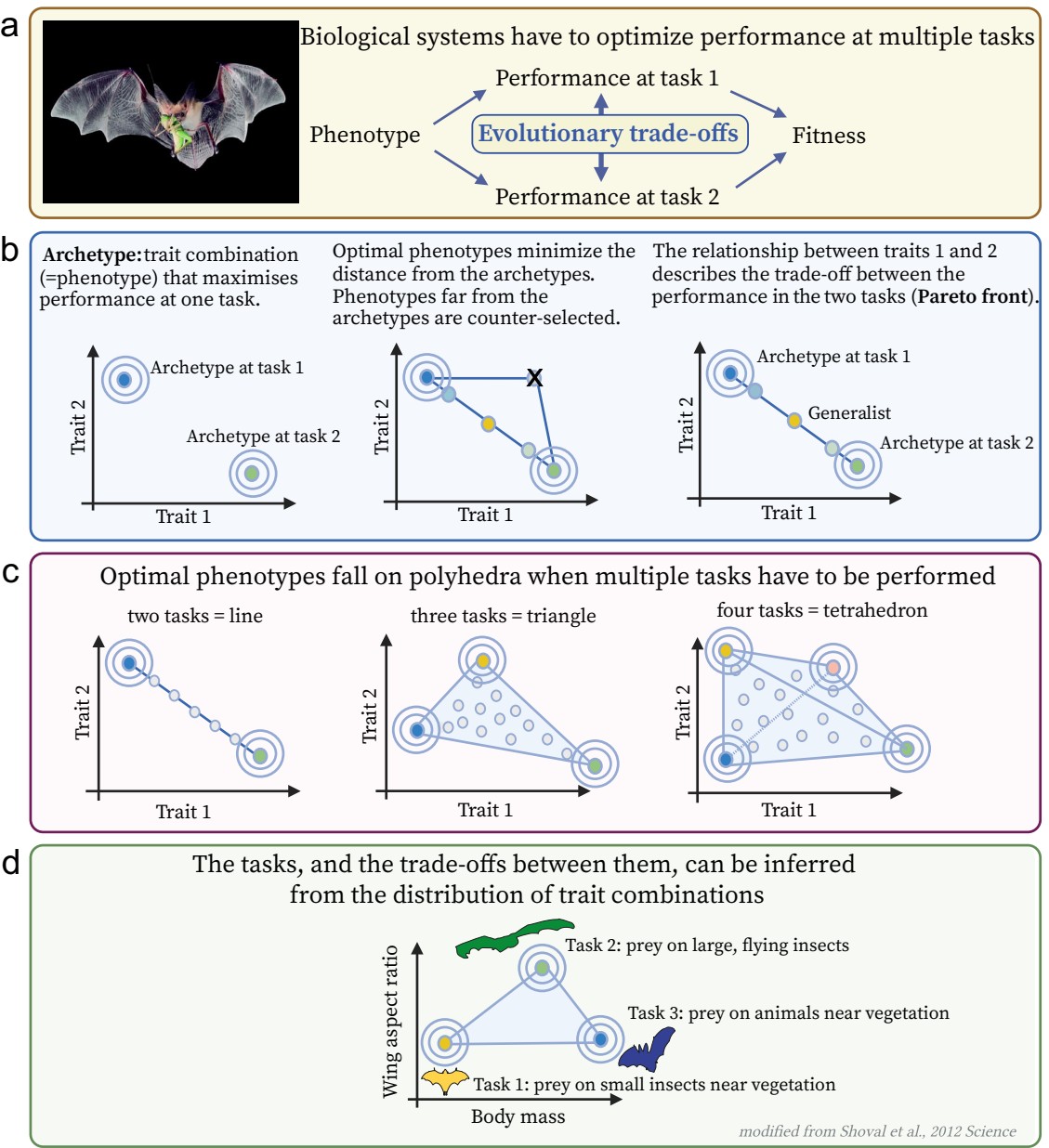

**Fig. 1 | The Pareto task inference framework. a** Biological systems perform multiple tasks to maximize their fitness function. Often, these tasks are in trade-off with each other, making it impossible for a species to optimize performance levels for all tasks simultaneously. The competition between tasks influences the selection of traits, leading organisms to strike a precise balance between traits to maximize their fitness function. **b** Pareto task inference is a general framework that predicts that tasks in trade-off lead to low-dimensional distributions of phenotypes, known as Pareto fronts, in the trait space. The vertices of the Pareto front are home to archetypal phenotypes, or specialists, which maximize performance at one specific task. The internal part of the Pareto front is occupied by generalists, which are good at performing all tasks but do not maximize performance in any specific task. **c** In the two-dimensional trait space, optimal phenotypes fall into convex polygons. For two tasks in trade-off, phenotypes fall into lines; for three tasks, they fall into triangles; for four tasks, they fall into polyhedrons, and so on. **d** An example of a triangular Pareto front in the trait space of body mass and wing aspect ratio in different species of bats. The three vertices correspond to archetypes specialized in different predation strategies (modified from ref. 50). Silhouette figures were contributed by various authors with a public domain license (public domain mark 1.0; CC0 1.0) from PhyloPic (http://phylopic.org). Source data are provided as a source data file.

the sum of squares error (Supplementary Fig. 1). Finally, we determined the position of the vertices using a hyperspectral unmixing algorithm, Sisal[62] (Supplementary Fig. 2), and the error of the vertex location was estimated using bootstrapping[47].

Altogether, these results indicate that tetrapods are enclosed within a triangular Pareto front in the double logarithmic space of BM and SL, as expected in the presence of trade-offs. A closer inspection of the geometry of BM-SL phenotypic distribution suggests that increasing body size is associated with a moderate increase in sperm length, as indicated by the side of the triangle connecting vertex T1 and

T3, which has a slightly positive slope. However, very long sperm evolved only in medium-sized tetrapods (Fig. 2a).

**Taxonomic distribution within the Pareto front**
The vertex T1 (green), which is characterized by species with the lowest body mass and shortest sperm length, is exclusively populated by amphibians. In contrast, the vertex T2 (grey), which represents species with the longest sperm and small-to-intermediate body mass, is populated by species from all classes except reptiles, with a higher frequency of mammals and birds (Fig. 2a). Mammals with intermediate

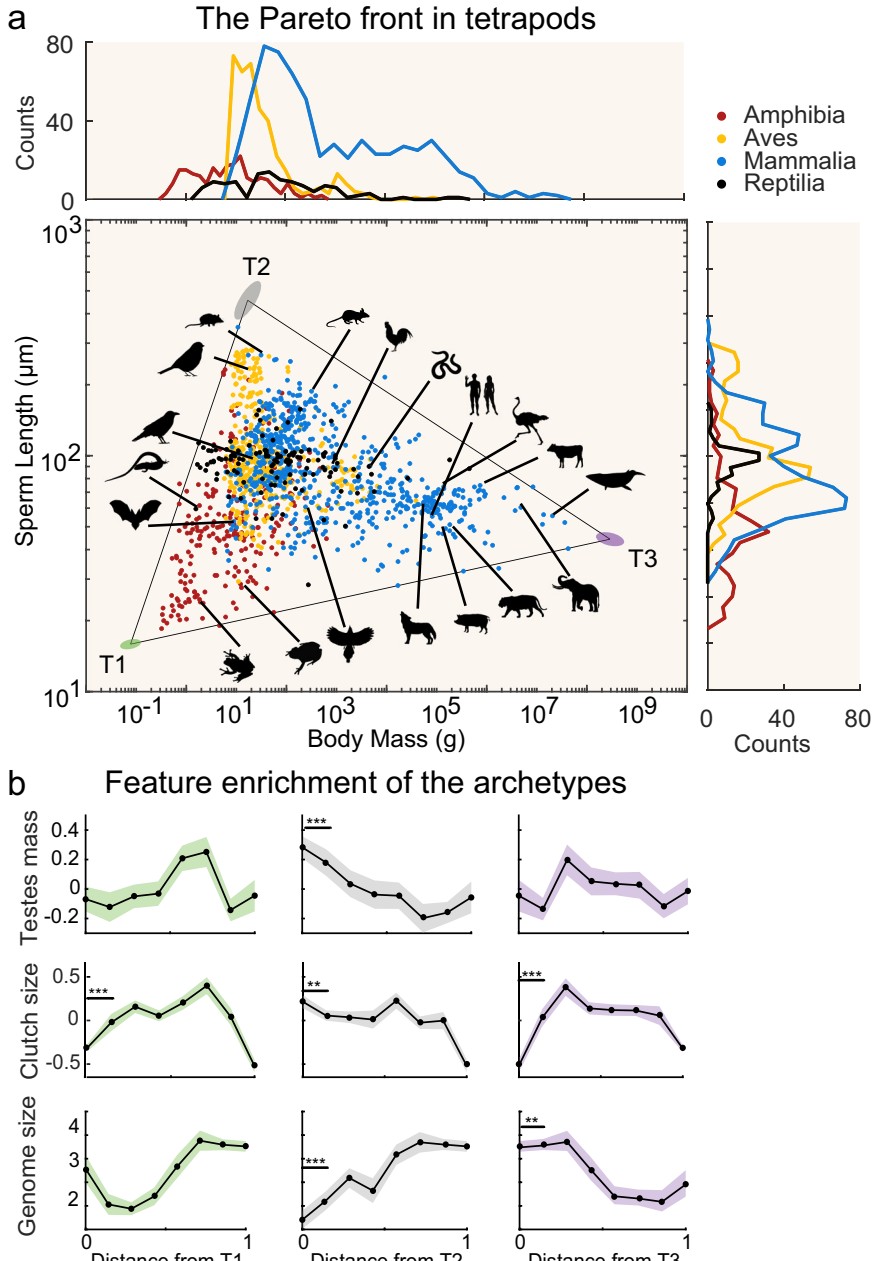

**Fig. 2 | The triangular Pareto front in the trait space of sperm length and body mass in tetrapods. a** We pooled together 1388 species from four classes (orange: Amphibia, green: Aves, blue: Mammalia, black: Reptilia). Individual species (dots) are plotted in the log-log space of body mass and sperm length. Some species are highlighted by silhouette figures as examples. Species are distributed within a triangular-shaped Pareto front (*t*-ratio test, *p* = 0.001). The three vertices of the triangle host the archetypal species: vertex T1 (green) is populated by frogs, vertex T2 (grey) is populated mainly by birds and small mammals, while vertex T3 (purple) is populated by large mammals. The ellipses represent the error in the positions of the vertices as determined by bootstrapping. Histograms show the species distributions in sperm length and body mass associated to each class. Silhouette figures were contributed by various authors with a public domain license (public domain mark 1.0; CC0 1.0) from PhyloPic (http://phylopic.org). **b** We computed the feature density of testes mass (residuals) (top panels), clutch size (within class *z*-scored values) (middle panels), and genome size (bottom panels) as a function of the Euclidean distance from the three vertices of the triangle. Species near vertex T2 show larger clutch size and testes mass, while species close to vertex T3 carry larger genome size. Solid lines represent the mean while shaded areas represent the standard error across species (number of species in each bin: clutch size = 146, testes mass = 120, genome size = 41). Statistical comparisons were performed using a two-sample two-sided *t*-test (**\*\*p* < 0.01, \*\*\*p < 0.001*). Source data are provided as a source data file.

sperm length and large body mass are located at the vertex T3 (purple). Finally, reptiles are located mainly in the middle of the Pareto front, although they show a relatively large range in both body mass and sperm length. The taxonomic richness of the vertices of the triangle may partly reflect the taxonomic sampling: vertices are less species-dense, and taxonomic groups with smaller sample sizes are less likely to be represented in the vertices. An alternative, not mutually exclusive explanation, is that factors influencing sperm length evolution in relation to body size are not homogenously distributed within the triangle.

### Clutch size, testes mass and genome size are enriched at the vertices of the Pareto front

The triangular distribution observed in the trait space of BM-SL suggests that trade-offs among tasks have constrained the evolution of sperm length in relation to body mass. To infer the

**Table 1 | A summary of the number of species for which sperm length, body mass, testes mass, clutch/litter size and genome size data were available**

|  | Sperm length and body mass | Testes mass | Clutch/litter size | Genome size |
|---|---|---|---|---|
| Amphibia | 231 | 197 (85%) | 182 (79%) | 36 (16%) |
| Aves | 399 | 324 (81%) | 370 (93%) | 111 (28%) |
| Mammalia | 643 | 408 (64%) | 522 (81%) | 168 (26%) |
| Reptilia | 115 | 29 (25%) | 92 (80%) | 16 (14%) |
| Total | 1388 | 958 (69%) | 1166 (84%) | 331 (24%) |

In brackets the proportion of species with testes mass, clutch/litter size, and genome size in relation to the number of sperm length and body mass data is reported.

competing tasks at play, we next explored traits enriched at the vertices of the triangular distribution[17]. According to Pareto Task Inference, the vertices of the triangle are home of the archetypes, i.e., species/phenotypes exhibiting the highest performance for a specific task (see "Methods" and Fig. 1). To this end, we considered three major factors proposed to influence the evolution of sperm length: sperm competition, clutch size, and genome size. We used the feature enrichment approach to determine whether these features exhibit maximal or minimal scores at the vertices of the triangle and decrease or increase as a function of the distance from the archetypes (see "Methods").

We examined whether relative testes mass (as a proxy for the level of sperm competition, see Supplementary Information, Supplementary Figs. 3 and 4), clutch size and genome size were enriched at the archetypes (Fig. 2b). Our dataset contains clutch size data for 1166 species, relative testes mass values for 958 species, and genome size data for 331 species (see Table 1). Clutch size is expected to relate to both sperm length and body mass and may be maximized under specific body mass–sperm length combinations. In particular, as clutch size is negatively associated with body mass in mammals and, to a lesser extent, in birds[32] and positively associated with sperm length in some taxa, e.g. ref. 63, we expected T2 to be enriched for large clutch size values. Relative testes mass is expected to be related to sperm length[9]. Moreover, genome size is also expected to be minimized at T1 and maximized at T3, reflecting the positive association with body mass. If, instead, genome size was the main driver of sperm length evolution, it should be maximized at T2. We found that, as predicted by theory and previous empirical evidence, testes mass and clutch size values are maximized at vertex T2 and decrease moving away from it (low relative testes mass and clutch size for species near vertices T1 and T3, see Fig. 2b). In partial contrast to predictions, genome size is maximized at the T3 vertex, corresponding to animals with large body mass, but minimized at the T2 vertex, corresponding to species with long sperm.

## Pareto front in relation to thermoregulation and fertilization modes

We next investigated the influence of thermoregulation and fertilization modes on shaping the triangular Pareto front in the trait space of sperm length and body mass. We thus performed two separate analyses by grouping the species based on (1) whether they were endotherms or ectotherms, and (2) their fertilization mode, i.e., being an external or internal fertilizer. We found that both endotherms and internal fertilizer species generate significant triangular Pareto fronts ($t$-ratio test, $p = 0.004$ and $p = 0.012$, respectively; see Fig. 3a, c), whereas ectotherms and external fertilizer species do not ($t$-ratio test, $p > 0.05$). We next performed feature enrichment analyses on the vertices of the significant Pareto triangles and found that, similar to the main triangular Pareto front, in both endotherms and internal fertilizers, clutch size is maximized at the vertex En2 and I2(Fig. 3b, d), which is populated by species with long sperm and small-to-intermediate

body masses. Our results suggest that the evolutionary constraints shaping the sperm length–body mass distribution in tetrapods may be conserved within two functional groups, namely endotherms, and internal fertilizers.

## Pareto front in the four classes of tetrapods

We then tested whether mammals, reptiles, birds, and amphibians formed significant Pareto fronts in the trait space of BM-SL (Fig. 4). We found that mammals and birds form significant triangular Pareto fronts ($t$-ratio test, $p = 0.002$ and $p = 0.007$, respectively; Fig. 4a, b), while amphibians form a marginally non-significant triangle ($t$-ratio test, $p = 0.104$, Fig. 4c), and reptiles do not form a significant Pareto front ($t$-ratio test, $p = 0.582$, Fig. 4d). Consistent with the case of endotherms and internal fertilizers, the vertex T2 in mammals also enriched in clutch size, but not in testes mass (Fig. 4f).

## Robustness of the Pareto front: control for phylogenetic bias

To address the potentially confounding effects of phylogeny when assessing the statistical significance of the triangular Pareto front, we employed the SibSwap approach[64]. This method acts as a more conservative test of triangularity by comparing the distribution of actual data with null distributions generated by randomly permuting the values of each trait independently (i.e., sperm length and body mass) within each set of terminal nodes sharing a parental node (hereinafter referred to as sibling tips). We first generated a temporally resolved phylogenetic tree from our dataset, following a previously described approach[1] (see "Methods", Fig. 5a). The tree consisted of 1345 terminal nodes, spanning from 355 million years ago (Mya) to the present. We set different time points, ranging from 355 to 20 Mya, and identified, the closest ancestor nodes for each time point. We then used the branch length information to group the sibling tips connected to the ancestor nodes at each time point. For instance, at ~255 Mya, the four sibling tips correspond to the four tetrapod classes (Fig. 5b, top panel). Next, we applied the SibSwap approach by randomly permuting the values of sperm length and body mass independently among species within each identified sibling tip (Fig. 5c). The SibSwap-shuffled datasets were obtained by shuffling the species within sibling tips, ensuring that the new sperm length-body mass associations preserve the phylogenetic dependencies. We then compared the $t$-ratio of the Pareto front obtained from the original dataset with those obtained from the SibSwap-shuffled datasets. If the Pareto front was affected by phylogenetic dependencies, we would expect no significant differences between the original and SibSwap-shuffled distributions ($p$ values > 0.05). Instead, significant differences ($p$ values < 0.05) between the original and the shuffled trait space distributions would indicate that the Pareto front is not affected by phylogenetic dependencies. Our results show that the triangular Pareto front remained robust to phylogenetic dependencies until approximately 65 Mya, at which point the $p$ value exceeded the 0.05 threshold (Fig. 5d).

We further analyzed phylogenetic dependencies from 65 Mya to the present by developing a modified SibSwap approach (see Supplementary Information for details and Supplementary Fig. 5), which involves averaging the values of SL and BM across species within each sibling tip. We then used the $t$-ratio test to statistically compare the distribution of these average values against null distributions and found that the Pareto front was robust against phylogenetic dependencies as a function of time points (Supplementary Fig. 5c). Importantly, the null distributions were generated by shuffling traits across all average values, treating them as independent data points in the trait space (see Supplementary Information). Our results indicate that, irrespective of whether considering older (from 355 Mya to 65 Mya) or more recent (below 65 Mya) phylogenetic relationships among species, no significant phylogenetic bias could impact the validity of the triangular Pareto front.

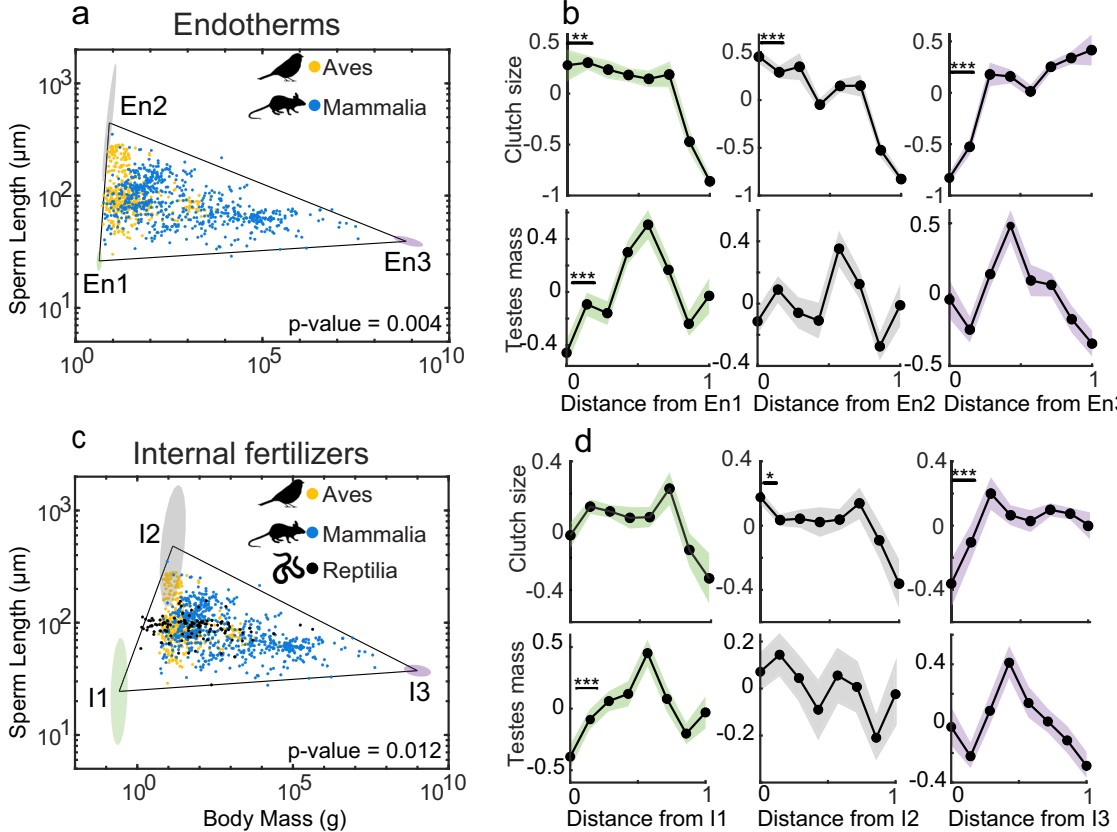

**Fig. 3 | Robustness of the Pareto front within functional groups. (a)** Endo-therms, i.e., Aves and Mammalia (N = 1042), and (**c**) internal fertilizers, i.e., Aves, Mammalia and Reptilia (N = 1157) show a triangular Pareto front (t-ratio test, p = 0.004 and p = 0.012, respectively). **b, d** We investigated the feature density on (**b**) Endotherms and (**d**) Internal fertilizers of testes mass, clutch/litter size and genome size. We reported statistically significant features that enriched at least one vertex of the triangles. Solid lines represent the mean while shaded areas represent the standard error across species (number of species in each bin: **b** clutch size = 112, testes mass = 92; **d** clutch size = 123, testes mass = 95). Statistical comparisons were performed using a two-sample t-test (*p < 0.05, **p < 0.01, ***p < 0.001). Silhouette figures were contributed by various authors with a public domain license (public domain mark 1.0; CC0 1.0) from PhyloPic (http://phylopic.org). Source data are provided as a source data file.

## Robustness of the Pareto front: finite-sampling bias

Given the unequal distribution of the number of species among the four classes of tetrapods in our dataset (46.3% mammals, 28.7% birds, 16.6% amphibians, 8.3% reptiles), we investigated the potential impact of sampling bias on the Pareto front at the class level. To mitigate the influence of numerical imbalances, we randomly selected 115 species per class, which corresponds to the maximum number of Reptilia in our dataset, for a total of 460 species (~33% of the data points). Following the SibSwap approach, we independently randomized the values of body mass and sperm length within each class. Our analyses revealed a significant triangular Pareto front (t-ratio test, p = 0.021, Fig. 6a), suggesting the robustness of the Pareto front against sampling bias at the class level, even when the dataset is reduced to approximately 33%. Further, we addressed sampling bias at the order level by considering only those orders with at least 30 species. This resulted in 9 out of 35 orders, for a total of 270 species (~20% of the data points). After shuffling the body mass and sperm length values across only the species within each order, the Pareto front remained marginally significant (t-ratio test, p ~ 0.05).

Additionally, we randomly selected subsamples of data points, ranging from 5 to 80% of the entire dataset, and for each, we conducted the triangularity test using the SibSwap method to account for the full phylogenetic dependencies. We found that when sampling over 30% of the dataset (~460 species) produced significant p values until ~90 Mya (Fig. 6b). In the range of 20–30% the p values were marginally significant, while below 20% the triangularity test was consistently non-significant (Fig. 6b). Overall, the Pareto front

identified in tetrapods is robust in the taxonomic groups of mammals and birds, and against phylogenetic dependencies across different time points along the phylogenetic tree. Importantly, the Pareto front in tetrapods remains robust whilst down sampling to one-third of the data points from our tetrapod dataset.

## Discussion

In this study, we postulated that the relationship between sperm length and body mass in tetrapods is shaped by trade-offs between multiple tasks. To test this hypothesis, we used the Pareto Task Inference (ParTI)[47], a multi-objective optimization algorithm that can be used to infer trade-offs among multiple tasks. We provide clear evidence that across tetrapods, sperm length variation in relation to body mass lies on a triangular distribution. According to the Pareto Task Inference, the occurrence of a triangular distribu-tion suggests that different interacting evolutionary forces are simultaneously acting on the relationship between sperm length and body mass. To infer which evolutionary factors may affect the sperm length-body mass phenotypic space, we explored the properties encoded by the species at the vertices of the Pareto front. We considered the three main factors that have been pro-posed to influence sperm length evolution, namely the level of sperm competition, the number of eggs per clutch, and the size of the genome[2]. Our results reveal that these traits are either maximal or minimal at the vertices of the triangular Pareto front, demon-strating that body mass is associated with sperm length mainly through sperm competition and clutch size in a non linear way.

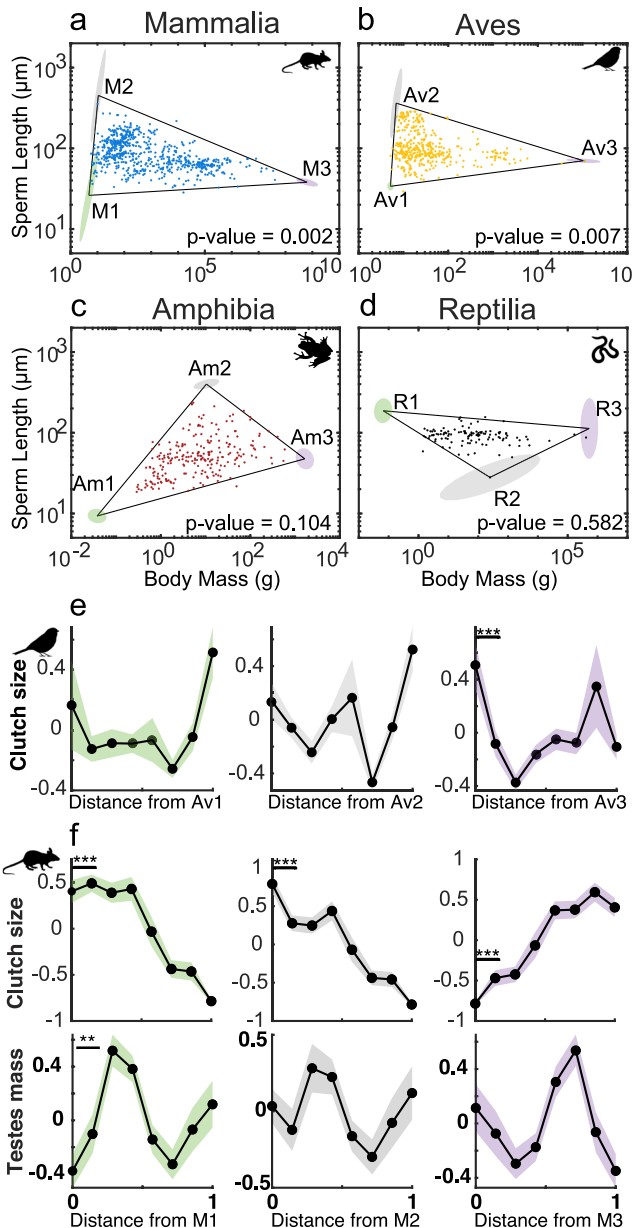

**Fig. 4 | Robustness of the Pareto front in the four classes of tetrapods.**
**a**, **b** Mammalia ($N = 643$), and Aves ($N = 399$) show a triangular Pareto front ($t$-ratio test, $p = 0.002$ and $p = 0.007$ respectively). **c**, **d** Amphibia ($N = 231$) and Reptilia ($N = 115$) do not fall within a triangular Pareto front ($t$-ratio test, $p = 0.104$ and $p = 0.582$ respectively). **e**, **f** We studied the feature density of clutch size and testes mass on Mammalia and Aves. We reported statistically significant features that enriched at least one vertex of the triangles. Solid lines represent the mean while shaded areas represent the standard error across species (number of species in each bin **e** clutch size = 46; **f** clutch size = 65, testes mass = 51). Statistical comparisons were performed using a two-sample $t$-test (**$p < 0.01$, ***$p < 0.001$). Silhouette figures were contributed by various authors with a public domain license (public domain mark 1.0; CC0 1.0) from PhyloPic (http://phylopic.org). Source data are provided as a source data file.

Our results unveil some important aspects of sperm length evolution in tetrapods. Firstly, we showed that sperm length variation is significantly associated with body mass, though not linearly. This may explain why many previous studies failed to find a significant association between sperm length and body mass, e.g., refs. 14,44,65. We demonstrated that in tetrapods sperm length progressively increases with body mass (positive slope of the triangle line connecting T1 and T3, representing small and large bodied-tetrapods). Most of the

variation in sperm length, however, is observed at intermediate body size, where very long sperm evolve in species with large clutch size and with high levels of sperm competition (T2). This result suggests that very large and very small tetrapods cannot evolve large clutches and intense sperm competition (represented by relatively larger testes). Instead, large clutches and intense sperm competition are associated with the evolution of long sperm in intermediate body-sized species (T2). Life-history trade-offs linking body size with reproductive strategy, in terms of clutch size and sperm competition, are therefore reflected in the triangular Pareto front distribution of BM-SL which is observed with striking regularity at the level of the entire group of tetrapods, and in most of their functional and taxonomic subgroups. We discuss these trade-offs below.

We found support for the key factors that have been postulated to impact sperm length evolution and demonstrated that these factors are linked non-linearly with body mass. First, confirming previous studies, we found that sperm length is maximized in species with intense sperm competition. Somewhat unexpectedly, we found that sperm competition is maximized only in species with intermediate body size (T2), whereas very small and very large tetrapods show a relatively low level of sperm competition (T1 and T3). The strength of the association between sperm competition and sperm length, therefore, does not seem to increase as a function of body mass as postulated by the dilution hypothesis[20]. The fact that sperm length evolution is constrained at the extremes of the body mass range in tetrapods (moving from T2 to T1/T3, body mass decreases/increases and sperm length decreases) may reflect the association between body size and ecological/social factors, such as population density, which affect male and female reproductive strategies and hence levels of sperm competition[18–21]. However, the observation that intense sperm competition is not observed also at extremely small body masses is unexpected, and we are not aware of any hypothesis that may explain why such reproductive strategy may be constrained in small bodied-species.

Our results show that clutch size is maximized in species with long sperm and intermediate body masses (T2), suggesting that clutch size is, along with sperm competition, the main driver of the evolution of long sperm in tetrapods. Surprisingly, this result also applies to frogs (which populated the area close to T1 and T2 ), in which previous works did not find an association between clutch size and sperm length, e.g., refs. 24,27. However, in many frogs sperm have to swim through a thick gelatinous coat before reaching the egg fertilization site, a process that may require many minutes[24] and sperm length is also positively influenced by egg size[27]. It is thus possible that large clutch sizes have favored the evolution of longer sperm that may be more efficient in swimming into a large clutch. Interestingly, we also found that both small (T1) and large (T3) species minimize clutch size, confirming that clutch size is associated to body mass variation, but differently from what was previously shown. The apparent constraint between small body mass and clutch size, however, is apparent when tetrapods are jointly analyzed. In endotherms (Fig. 3a), in which the minimum body mass is larger than in ectotherms, small species capture all the variation in sperm length, and the two vertices along the $y$-axis (T1 and T2) significantly enrich for large clutch sizes. This suggests that only extremely small ectotherm tetrapods are limited in the evolution of their clutch size (and hence of their sperm length). Accordingly, a previous work investigating the relation between body mass and longevity in endotherms using the Pareto framework[17] also found that small endotherms can both maximize and minimize clutch sizes. Overall, we show that clutch size does not linearly correlate with body mass, highlighting the power of Pareto Task Inference in detecting non-linear associations between traits.

We found that the size of the genome is maximized at the T3 vertex, which corresponds to animals with low-to-intermediate sperm lengths and large body masses, and it is minimized in species with long

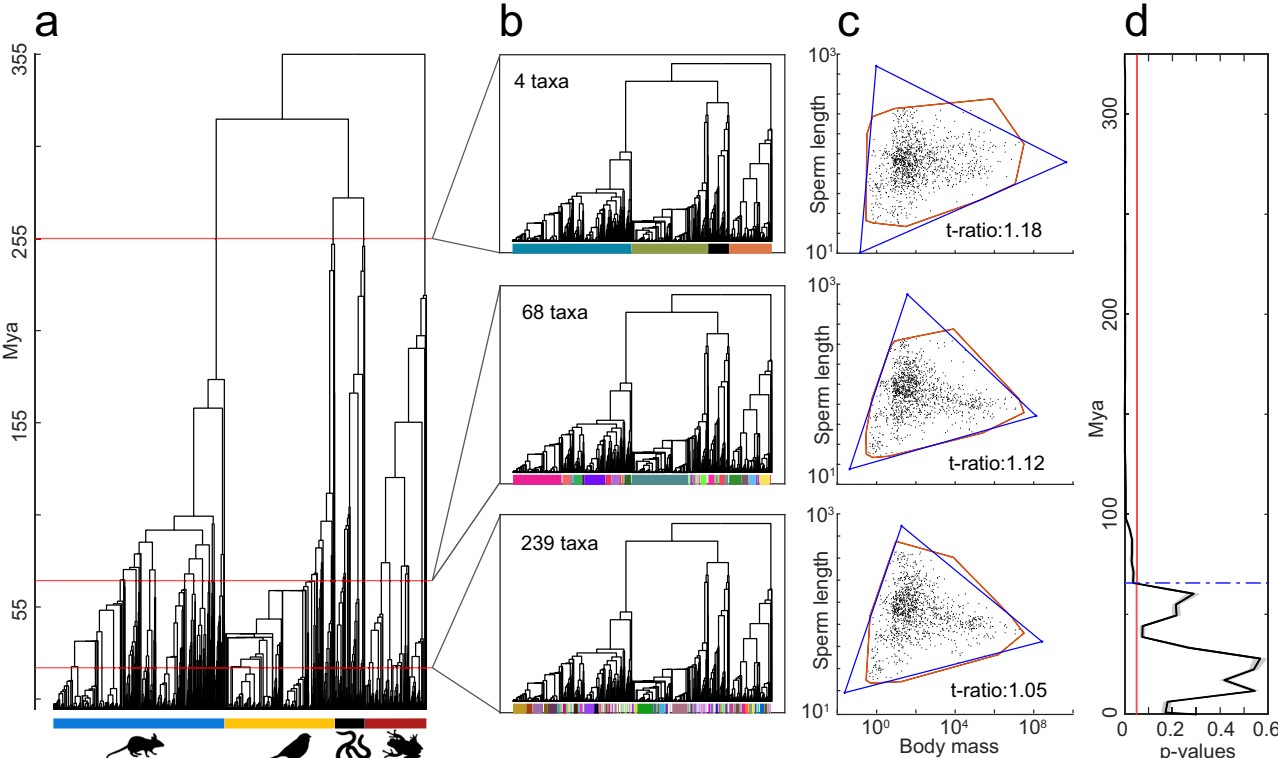

**Fig. 5 | Robustness of the Pareto front: control for phylogenetic bias. a** We first built the time-resolved phylogenetic tree on 1347 out of 1388 tetrapods using the approach in Kahrl et al.[1]. The four-colored bars indicate the four classes of Mammalia (blue), Aves (yellow), Reptilia (black) and Amphibia (red). We used several time points along the phylogenetic tree (e.g. see the horizontal red lines for some examples) to determine the sibling tips (i.e., species characterized by an identified common ancestor). **b** Three examples of tree sibling tip classification based on common ancestor nodes identified using three different time points. **c** Examples of SibSwap-randomized distributions of species in the trait space of body mass and sperm length. Each randomized distribution was generated using the SibSwap approach, which is based on randomizing separately the values of body mass and

sperm length among the species within the same sibling tip. The *t*-ratio defines the ratio between the polytope and the convex hull (see "Methods"). **d** We computed the *p* values based on the *t*-ratio test using the SibSwap randomization approach at different time points. The solid line and the shaded area represent the mean and standard deviation across 100 iterations respectively. The vertical red line corresponds to *p* value = 0.05, while the blue dashed line marks the limit between not significant phylogenetic dependencies (until 65 Mya) and significant phylogenetic dependencies (from 65 Mya to the present). Statistical comparisons were performed using a one-sample one-sided *t*-test. Silhouette figures were contributed by various authors with a public domain license (public domain mark 1.0; CC0 1.0) from PhyloPic (http://phylopic.org). Source data are provided as a source data file.

sperm and intermediate body masses (T2). The positive correlation between body mass and genome size has been shown in previous comparative analyses and is not surprising[33,34]. The characterization of the T3 archetype may be congruent with the prediction that large genomes are associated to longer sperm, according to the hypothesis that larger genomes require longer sperm flagella to counteract the drag associated with a larger sperm head. However, this hypothesis is clearly contradicted by the observation that the archetype associated with very long sperm (T2) is characterized by smaller-than-average genome size, the opposite of what we initially predicted. We can speculate on the reason why species with very long sperm have smaller genomes. One possible explanation is that a large genome size is negatively correlated with meiosis duration[66] and species with very long sperm and large testes (T2) are characterized by a high rate of sperm production[67], which may be incompatible with large genomes.

Finally, we found that a triangular Pareto front in the BM-SL trait space is maintained within subgroups of tetrapods, namely mammals, birds, endothermic species, and internal fertilizers, suggesting that similar evolutionary pressures characterize the evolution of sperm length in relation to body mass within phylogenetic and functional subgroups within tetrapods. Interestingly, while the maximum sperm length is never observed in the species at the two extreme ranges of body mass variation, the location of the T2 vertex (species with long sperm) in relation to body mass variation differs among groups. For example, T2 is located very close to the lower extreme of body masses

in birds and mammals (endotherms) and around intermediate body mass values in the amphibians, possibly reflecting differences in the way the trade-offs linking BM and SL evolution are modulated in these groups. Irrespective of the interpretation of the evolutionary trade-offs that may underly these observed patterns, the results highlight the power of our statistical framework to detect complex, non-linear associations between traits that may be overlooked when traditional phylogenetic regression methods are employed. Unlike comparative analyses that typically control for body size variation[68], the Pareto Task Inference framework allows to investigate the evolution of sexual traits by explicitly including body size variation. We demonstrated that including body size variation can unveil previously unknown trade-offs among different factors, which can be inferred from the distribution of species in the trait space.

However, we also acknowledge some limitations of our statistical approach. Firstly, the method is data hungry, and requires larger sample sizes than comparative analyses to explore the entire trait space determined by the evolutionary processes of interest. When the number of species is low, statistical power to detect Pareto fronts may be insufficient, as in the case of amphibians and reptiles, where Pareto fronts were not significant probably due to the small sample size (231 and 115 respectively), while in the case of mammals and birds, we found significant triangular Pareto fronts (643 and 399 species respectively). In line with this observation, we found that the minimum number of tetrapod species required to obtain a significant triangular Pareto front

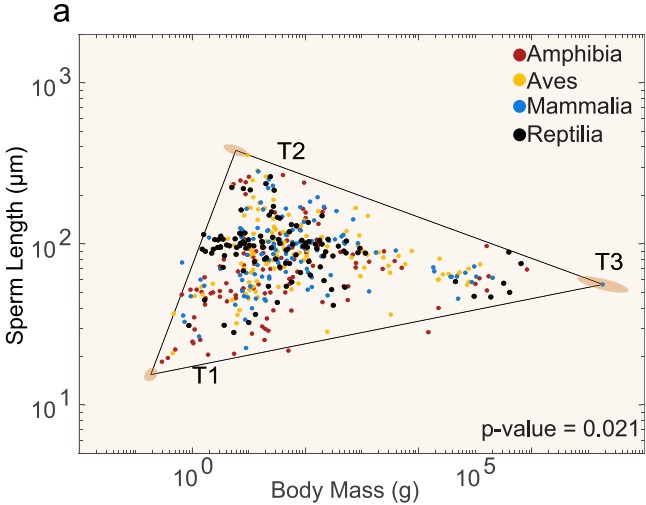

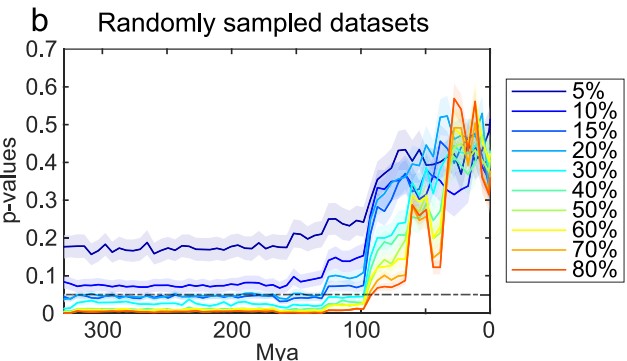

**Fig. 6 | Robustness of the Pareto front: control for finite-sampling bias.**
**a** To have the same number of species for each class, we subsampled the initial dataset of 1388 species by randomly selecting 115 species from each class, which resulted in a total of 460 species. We then tested the triangularity of such distribution against randomized distributions following the SibSwap approach[64]. Data fall within a significant triangular Pareto front (*t*-ratio test, *p* = 0.021). **b** We performed the test of triangularity by randomly selecting between 5 and 80% of the initial dataset. Using the SibSwap approach, we tested for the phylogenetic dependence on each subsampled dataset. Solid lines represent the mean while shaded areas represent the standard error across 30 iterations. Source data are provided as a source data file.

is 450, which correspond to ~30% of the whole dataset. Secondly, while this approach is useful for revealing previously unknown or untested trade-offs, it does not allow to demonstrate causal relationships between the traits enriching the vertices and the traits determining the Pareto front. For example, in our study, although testes mass and clutch size are both maximized in species with long sperm and intermediate small-to-intermediate body mass, our analysis does not allow us to disentangle whether large clutches have favored the evolution of polyandry and longer sperm evolve in response to the associated increased level of sperm competition, or if instead long sperm have a fertilization advantage when clutches are large, and large clutches are independently associated with polyandry. A similar reasoning could be applied to the negative correlation between genome size and sperm length. Genome size could be constrained by the evolution of long sperm or by the evolution of large clutches, which covary with sperm length.

In conclusion, we present a comprehensive analysis of sperm length variation across tetrapods, based on the Pareto Task Inference framework. All the hypotheses on the evolution on sperm length proposed so far assume that factors determining sperm length evolution are mediated, directly or indirectly, by body size, although

empirical evidence was largely missing. Congruently, our analysis demonstrated the existence of a triangular Pareto front in the space of body mass and sperm length, suggesting that these two traits reflect underlying evolutionary trade-offs and constraints influencing sperm length evolution. We showed that long sperm evolve in species with intense sperm competition, as expected[9], and with large clutch size, as previously shown just in fishes[63]. We demonstrated that the evolution of long sperm is constrained in very large and very small tetrapods, and this pattern is confirmed also across taxonomic and functional subgroups. The reasons why large clutch size and high levels of sperm competition are not observed in very small and very large tetrapods probably depend on different ecological and morphological constraints which need to be further investigated. We also demonstrated that long sperm do not occur in species with large genomes, suggesting that genome size is constrained in long sperm species, rather than species with large genomes being constrained to produce long sperm, contrary to predictions[40,41].

## Methods
### Data collection
For each species we collected a maximum of 5 trait values, namely body mass, sperm length, clutch size, testes mass, and genome size, that we defined as vectors $\nu_i$ with $i = 1, \ldots, 5$. We collected sperm length (μm) data from the most comprehensive published dataset to date on sperm length[60] and from other sources, e.g., refs. 11,14,24. Body mass, clutch size, testes mass as a proxy for sperm competition[9], and genome size were collected from published and publicly available sources, e.g., refs. 69–73. We included only a single source per species; when more than one value for a single species was present, we prioritized the most recent one. More details on our data collection procedure can be found in the Supplementary Information and sample sizes for each trait are reported in Table 1. Body mass and sperm length were log10-transformed before analysis. As a proxy for sperm competition, we calculated the relative testes mass by obtaining within-class residuals from log-log linear regressions between testes and body mass (see Supplementary Information, Supplementary Figs. 3 and 4). We normalized clutch/litter size among classes by calculating the within-class *z*-scores.

### Pareto task inference
Each species' performance functions ($P_\alpha(\mathbf{v})$) depend on the specific combination of traits $\mathbf{v}$ for all tasks in trade-off ($\alpha = 1, \ldots, k$ tasks). A basic assumption for the Pareto theory is the existence of a general fitness function, which is an increasing function of all the performance functions $F(P_1(\mathbf{v}), \ldots, P_k(\mathbf{v}))$. Archetypes are the optimal phenotypes ($\mathbf{v}^\alpha$) whose performance function $P_\alpha(\mathbf{v}^\alpha)$ is highest at a specific vertex $\alpha$, which decreases as a function of distance from the vertex as $P_\alpha(\mathbf{v}) = \hat{P}_\alpha\left((\mathbf{v} - \mathbf{v}^\alpha)^\mathrm{T} \mathbf{M}(\mathbf{v} - \mathbf{v}^\alpha)\right)$, where $\alpha = 1, \ldots, k$ and $\mathbf{M}$ is a positive-definite matrix. As a result, no single species can excel at all tasks at the same time and the spatial coordinates of each species within the Pareto front provide information about the set of traits that will result in optimal Pareto solutions at any point along the fronts (Fig. 1).

### The *t*-ratio test to assess the statistical significance of the Pareto front
A common approach used in Pareto analyses to assess the statistical significance of fitting Pareto fronts to data points is to compute the *p* values using the *t*-ratio test[47,48,58]. The *t*-ratio defines the fraction between the area of the best-fitted polytope, computed through the Sisal algorithm[62], and the area of the convex hull that encapsulates the data points, computed through the "convhulln" algorithm in Matlab. If the distribution of data points in the trait space is triangular, the convex hull will resemble a triangular distribution, and its area will almost entirely fill the space within the triangular polytope that can be fitted on the data points. This means that the areas of the fitted

triangular polytope and the convex hull will be similar, and the *t*-ratio will be close to one. In our dataset, we obtained a *t*-ratio of 1.03 (see Supplementary Fig. 2a). Conversely, randomly distributed data points will have a cloud-like shape in the trait space. This cloud of randomized data points will still be fitted by a triangle, but in this case the convex hull will hardly occupy the space within the best-fitting triangular polytope, and the regions in space close to the vertices of the triangle will remain mostly empty. In this case, the *t*-ratios will be consistently larger than one. For instance, by randomizing the data points of our dataset we found a *t*-ratio of ~1.25 (see Supplementary Fig. 2b). Next, the *p* values are defined as the proportion of instances where the *t*-ratios of the randomized datasets are lower than those of the original dataset, divided by the total number of randomized datasets. We considered as statistically significant *p* values those that scored under 5% of times ($p < 0.05$).

### Randomized datasets in the trait space

In cases when the data points were considered as independent, the trait randomization was done by independently shuffling the values of sperm length and body mass across all points in the distribution. However, when phylogenetic correlations among data points became relevant, we followed the SibSwap randomization approach as described in ref. 64. This method proposes to randomly permute the values of each trait independently (i.e., sperm length and body mass) within each set of terminal nodes sharing a parental node. This approach preserves both the phylogenetic constraints and the marginal distributions of each trait[64].

### Feature enrichment of the archetypes

According to the Pareto Task Inference, archetypal species near the vertices should exhibit maximum values in a set of enriched features, while species far from the vertices should have lower score values in those features. The scores of such features should monotonically decrease with the Euclidean distance from the vertices. The features that we considered for this analysis were testes mass, clutch/litter size and genome size. To investigate which feature enriches at the vertices of the triangle, we computed the density profile of each feature in terms of mean values as a function of the Euclidean distance from the archetypes. To do so, we first sorted data points using their Euclidean coordinates in trait space from the given vertex. We then divided the distribution of species in the trait space into 8 equally populated bins. In each bin, we computed, for the given feature, the mean value of that feature across the species within the given bin. We then defined as enriched features those that passed the *p* value test based on the two-sample *t*-test computed between the scores of the feature at the bin nearest to the vertex and the scores of the rest of the distribution of points excluding the species of the nearest bin.

### Phylogenetic tree building

We generated the phylogenetic tree following the approach outlined in ref. 1. Briefly, we used the list of species in our dataset to generate a tree using Open Tree of Life v.12.3[74] (http://opentreeoflife.org/) with the package rtol[75] and we matched the names using the tnrs_match_names and tol_induced_subtree functions. To time calibrating the phylogenetic tree we used the function congruify.phylo[76] within the PATHd8 scaling algorithm[77] from the package geiger v. 2.0.6[78]. The function uses the information from TimeTree.org[79] to add branch length information along the entire phylogenetic tree. Finally, polytomies were randomly solved using the function multi2di from the package ape[80]. The final sample size of our phylogenetic tree comprised 1347 species.

### Reporting summary

Further information on research design is available in the Nature Portfolio Reporting Summary linked to this article.

## Data availability

All data and the associated references are reported in the dataset uploaded on Figshare (https://figshare.com/articles/dataset/Dataset_Tetrapod_sperm_length_evolution_in_relation_to_body_mass_is_shaped_by_multiple_trade-offs_/26022289). Source data used for generating each figure are provided as a Source Data file. Source data are provided with this paper.

## Code availability

Pareto analyses were performed in MATLAB (version 2021a), while the tree building was performed in R (version 4.3.2). We provide a customized MATLAB code on GitHub repository to carry out all Pareto analyses and generate the results presented in the present work (https://github.com/lorenkocillari/Pareto_Sperm_Length_Evolution)[81].

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

## Acknowledgements
We thank Ariel Kahrl for providing support in the phylogenetic tree building and Bernd Serf (CF Life Science Computing at the FLI) for the IT support. We thank Samuel Allan Kean and Carolina Zucchi for proofreading the article and Beniamino Tuliozi for comments on an early draft of the manuscript. We would also like to thank all the people that over the past decades collected the data that we used in this study and made it publicly available to the community. S.C. was supported by a grant from the University of Padova (BIRD-175144-2017 to A.P.) and by the Valenzano lab core budget.

## Author contributions
M.B.R., A.M. and A.P. conceived the study, S.C. and A.P. collected the data, L.K. performed the analyses and produced the results, L.K. and S.C. led the writing, A.P., M.B.R., A.M. and F.S. contributed to the revisions of the original draft. All authors approved the final version of the manuscript.

## Funding

## Competing interests
The authors declare no competing interests.
