## [Peer Review File · Nature Communications]

REVIEWER COMMENTS

Reviewer #1 (Remarks to the Author):

In this manuscript, Koçillari and colleagues tackle a big question using an innovative (to the field) approach applied across tetrapods. Specifically, the authors set out to explain the tremendous variation observed in sperm morphology and how this variation is shaped by body size. In addition, they assess other factors which are argued to influence sperm size evolution, including sperm competition risk, clutch size and genome size. The use of a Pareto Optimality approach to examine the evolution of reproductive traits is a novel addition to the field and something that I had to invest some time to read up on. The Pareto Optimality approach is particularly well suited for uncovering complex trade-offs among competing traits that constrain how traits move in trait morphospace. While the applicability of the Pareto Optimality approach to address biological questions has been highlighted for a little over a decade (at least if we take the Shoval et al. 2012 Science paper as the starting point), I'm not aware of this approach being applied to sperm evolution. The authors have done a great job in identifying the potential of this approach and applying it to sperm trait evolution, where trade-offs are expected to influence the evolution of trait values.

Using this approach, the authors demonstrate that the morphospace when assessing sperm length and body size is a triangle, which is indicative of a trade-off between multiple tasks with three points of optimality (archetypes). The authors then explore if this pattern is observed when contrasting species with different fertilization modes, thermoregulatory methods, and when assessing different tetrapod classes. Together, this is an interesting set of analyses and helps to make sense of the seemingly unclear pattern that emerges when plotting sperm length and body mass.

Having suggested using Pareto plots that sperm length is likely subject to trade-offs is interesting but not entirely surprising – this is a common view but providing a robust demonstration is good to see. However, to further advance the field one would need to identify the traits driving these potential trade-offs. Here is where the manuscript fell short for me.

The treatment of the literature and motivation for the factors being tested is hard to follow and incorrect at times (lines 70-86 in particular and throughout the manuscript). The manuscript identifies several factors that may influence the body size-sperm size relation, most of which I had an issue with. These include:

1. Mass-specific metabolic rate. This point has been made elsewhere, but it is challenging to disentangle MSMR with dilution effects since these two factors, both of which have been hypothesized to explain

sperm length variation, covary with one another. However, after its introduction, MSMR was not included in the analyses, which was odd.

2. Clutch/litter size: The argument that clutch/litter size is positively correlated with sperm size in internal fertilizers is not supported by the citations provided. Similarly, in external fertilizers, the claim that larger clutches favor sperm number at the cost of size is not supported by the citations (also, your Stockley citation here should probably be Stockley et al. 1996, not 1997 as the latter paper doesn't look at female traits at all). Importantly, Stockley et al. 1996 shows a positive correlation between egg number and sperm length, which is contrary to the claims used to build the argument here. Most of the cited literature doesn't even mention clutch/litter size, which I found confusing. Overall, I'm not convinced there is good support at all for a clear link between clutch/litter size and sperm length (there is no correlation in most studies that have looked for a link, e.g., Byrne et al. 2003 Proc B). Sperm number is a different story – here there is evidence that larger clutches are correlated with larger testes but that is not the argument being advanced. Overall, this hypothesis lacked a clear connection to the literature and was poorly motivated, which left me wondering why it was included.

3. Genome size: I regard this as a 'zombie hypothesis', one that one die despite a general lack of evidence! It seems compelling to argue that there should be a link between genome size and sperm length, and there is plenty of genome data to bring to bear on a large comparative analysis like this. But is there any evidence that sperm size is related with genome size? In mammals, some early work examining sperm evolution in a broad comparative context by Gage (1998, Proc B, 265:97-103) found no association between sperm head length or head area and chromosome number or genome mass. Similarly, in ferns, spore size is not related with genome size (Dyer et al. 2013, BMC Plant Biol, 13:219). Instead, genome size appears to be more often related with egg size (e.g. in fishes, Hardie and Hebert 2004, Can J Fish Aquat Sci, 61:1636-1646; in snapping shrimp, Hultgren et al. 2021, J Evol Biol, 34:1827-1839).

4. Sperm competition: I agree with this argument.

My take away from these hypotheses was that they lacked a clear motivation for why they would drive trade-offs in sperm length with body size. Without the authors can expand on their motivation it remains unconvincing to me. An alternative interpretation to their results are that larger species have larger genomes and that is why genome size is closer to T3, and testes mass is positively correlated with clutch size and sperm size and that is why these traits are maximised at T2. Under this alternative interpretation it isn't clear what is driving the trade-offs but instead you are detecting multicollinearity effects among predictors variables. If this thinking is wrong headed (which is entirely possible as I'm new to this form of analysis) then more explanation is warranted.

Finally, my understanding is that the Pareto Optimality approach emerged from engineering and economics – two fields that usually don't take into account confounding effects of evolutionary history. Therefore, it wasn't clear to me how much the shared evolutionary history of the species/clades examined will influence the results. The authors do attempt to control for phylogenetic bias with a randomization test. However, this is a fairly weak approach to correct for phylogenetic effect given that it was performed at the class and order level and that it was performed in the absence of branch length information. Did you consider using phylogenetic independent contrasts for each trait value and using these in the analyses? Without a more robust account for the evolutionary history of the species examined it is challenging to interpret the findings. I recognize that in Shoval et al.'s 2012 paper two of their examples are comparative (Darwin's finches and bats – the latter of which is shown in your Figure 1). However, the lack of formal analyses of these data in Shoval et al. foreshadows the challenges of dealing with phylogenetic data using the Pareto Optimality approach that I'm not convinced has been addressed yet.

Specific comments:

Line 160: instead of 'population' do you mean 'species' here?

Reviewer #2 (Remarks to the Author):

This paper studies the tradeoffs in sperm

evolution. To do so it applies

Pareto task inference theory and uncovers evidence for three sperm tasks in tetrapods.

The paper is elegant and comprehensive.

I support publication of this paper, once the following suggestions for improvement are considered:

1. To shed further light on the tasks, please compare to previous work on three archetype life strategies in endotherms - shrew, bat and whale. To do so It would be important to show data when restricted only to endotherms. Then to compare to the mass -longevity triangle of Szekely 2016 . This paper provides additional life history traits that peak near each archetype. The bat archetype - protected niche including naked mole rat, primates etc, show . Small body size, large brain, small clutch size and large inter birth periods. It seems that one of the sperm archetypes is close to bats, and hence may be associated with the protected niche life strategy.

2. I am impressed by use of state of the art phylogenetic test for ParTI- better to use the original name "sibswap". It would be good to add an additional test for mammals alone while flipping the closets species on the tree.
3. It would help with skeptics to be more careful in the intro- the triangle is not proof of tradeoffs, it is the trait space distribution expected if there are three tradeoffs.
4. Pareto optimally theory is usually based on plotting in a space whose axes are performances at known tasks. Here the authors plot data in trait space and infer the tasks. It's better to call this "Pareto task inference theory". It's like an inverse problem on classical Pareto optimality theory.

Reviewer #3 (Remarks to the Author):

This study explored across tetrapods how body size variation could influence sperm length evolution via trade-offs between multiple variables such as sperm competition levels, clutch/litter size and genome size. To this end, the authors used Pareto optimality theory, which allowed them to consider multiple trade-offs simultaneously. They found the effects of body size on sperm length to be mediated primarily by sperm competition levels and clutch size.

Overall, this is a great way of dealing with broad and complex patterns of trait evolution, considering the oftentimes multivariate effects instead of singling out specific factors while ignoring others. The paper is timely and relevant in this regard. However, several important issues currently limit my enthusiasm for this study, as it is difficult to judge the correspondence between results and conclusions. These issues would need to be addressed before I can fully support the paper.

1. I find the current use of residuals very problematic for clutch size and testis mass (particularly the former). First, I am concerned about controlling for body size in these variables and then combining their residuals with absolute body size again in the same analyses. At least this is how I understand from the description that the analysis was done. If does not reflect the actual analysis, it may not be clear for other readers either, and the paper would benefit from a more detailed and careful description.

Second, even more importantly, I am concerned about combining the taxon-specific residuals in inter-taxonomic analyses. This approach essentially assumes that a value of 0.5 or -1.2 means the same in amphibians, birds or any other taxon. However, residuals very much depend on the distribution of the data and where the regression line falls. A poor correlation would result in a wide spread of data points and an unreliable regression line (and hence residuals), whereas a tight association would generate in a narrow distribution or residuals.

So, unlike sperm length or body mass, each residual value depends on the distribution of the remaining species of the taxon, and so values are more difficult to compare between taxa. Further, regression lines can be sensitive to outliers, which then translates into shifts in residuals across the dataset. Even standardizing residuals may not necessarily resolve some of these issues.

At this point, were residuals calculated for each given dataset (e.g. where subsamples were taken) or were those of the original dataset used for all analyses, regardless of which species were included? This is important to know precisely because of the dependency of residuals on the given data distribution.

2. The authors mention that they shuffled closely related species along the phylogeny to separate phylogenetic dependency. However, throughout the paper I found no mention of what phylogenetic trees were used to map the species on in the first place, nor how phylogenetic associations were accounted for in their 'original' analysis that the randomizations were compared to. If it is true that no formal phylogenetic control was undertaken, this would be very problematic and, quite frankly, inadequate. If phylogenetic covariance cannot be incorporated into the analytical approaches taken (e.g. as a correlation structure), could the authors use the phylogenetically independent contrasts instead, or some other form of phylogenetic correction?

3. I often found the introduction and discussion somewhat confusing as the terminology, theory or patterns were freely moved between taxa to back up arguments, even though the authors themselves acknowledged that their biology can differ in important ways. Please double-check carefully where such argumentation is appropriate and where it might be too much of a stretch. In general, the discussion as a whole could be more streamlined and more on-point. My specific comments below point out some of these cases.

Specific comments:

L24 "these factors" is unclear as it is too far removed from what it refers to. Consider reordering your statements.

L31 Since 'triangular' already means 'shaped like a triangle', so 'triangular-shaped' seems redundant (throughout the paper)

L53 shortest instead of smallest? Small and long use different dimensions and so are not direct contrasts.

L61 It is RELATIVE testis mass what is commonly used. This is important given the variation in body mass between species.

L66-67 In the same context, it might be important to consider the risk of sperm dilution as a constraint on sperm length that is less present in small species.

L80 though it seems to depend on the taxon according to Liao et al. 2018 (your ref. no. 51)

L99 remove redundant “been”

L210 I am not convinced that the claim “our results are not affected by the phylogenetic dependencies” is correct. The phylogenetic effect may simply not have been strong enough to overpower the Pareto front (i.e. being weak), but it does not mean (in my opinion) that there was no phylogenetic dependency whatsoever. That said, I also do not understand where phylogenetic trees were even incorporated into the analyses.

L220 The Methods section is in an odd place. To my knowledge, this usually comes after the Discussion in Nature journals. Where it is now, it interrupts the flow of the paper.

L222 I would consistently call this ‘sperm length’ (which it actually is), as sperm size sounds more like a 3-dimensional trait.

L251 What do you mean by “phylogenetically close?” Did you shuffle species within genera to compare the shuffled with the original data within a class? Or did you shuffle species between orders to do so or just randomly between all species of a class? This is not clear at all.

That said, please also note that the dataset contains crocodiles that would be more closely related to birds than to lizards and snakes... Given these are only three species, I am not worried that the outcome would change much, but it puts ‘closely related’ into perspective and how vague it is.

Finally, I see no mention of the phylogenies that were used and how the ‘original dataset’ was controlled for phylogeny, or how exactly species were shuffled across the phylogeny. If there was no phylogeny in the first place, how exactly did this shuffling address the phylogenetic dependency?

L253-254 I am not clear about this definition of the p-value. If p represents the proportion of t ratios that are lower than the original (i.e. farther away from capturing

L258 It would be helpful to provide at least a minimal description of the methods rather than only referring to a separate paper. Readers should not be asked to search for papers (possibly even behind a paywall) to get at least an inkling of what was done.

L276-315 This is a very long paragraph that could easily be topically split into two or three paragraphs for easier reading.

L282 “while it is”

L283 no hyphenation in “small or large body sizes”

L284 insert ‘the’ after ‘that’

L288 delete ‘into’ after ‘entering’

L289 I am not aware of either of the two cited papers reporting an association between the size of female sperm-storage organs and clutch size. Rather, a relationship was found between clutch size and the duration of sperm storage (i.e., essentially the laying period), which makes sense if one egg is laid per day in most species.

L289-292 This is a confusing sentence. First, it moves the discussion from interspecific relationships to the species level; second, it uses mammals with single offspring to back up the argument about body size, longevity and clutch size in birds. This line of argument is confusing: The association between body size and longevity is more likely to hold between than within species, as is the r vs. K reproductive strategy. Further, single offspring in mammals do not inform about clutch size in birds. Why not use the many long-lived birds (e.g. seabirds) that produce only 1-2 eggs per season (or possibly even only every other year)? Also, what seems entirely neglected in this line of argumentation is that parental care plays an important role (so it is not necessarily about longevity per se). Please also note that none of these animals produce only a single offspring – they just do so per reproductive bout. Finally, mixing ‘clutch’ and ‘mammals’ as if mammals also had clutches (rather than e.g. litters) further adds to the confusing argument. Please revise.

L292-294 Curiously, this argumentation seems to apply to fish but not amphibians based on the very paper cited here. Hence, the result in amphibians may have need another explanation?

L295 Note that sperm competition itself is not high or low, but its level can be. Alternatively, just say “strong/intense sperm competition” (or similar).

L307 It is not clear why this negative result is particularly interesting given there has not been much evidence for it in previous studies already? Also, it is not clear how the conclusion about genome size being constrained in species with high sperm competition levels is derived from sperm length and body mass variation (previous sentence)? Please be more specific about how these different traits are linked.

L344 This sentence is confusing. Maybe a comma is needed after clutch size to separate those parts of the sentence? Does this then express what you intended?

Supplementary Fig. 3: This is an unusual representation of the relationship between body mass and clutch size/testis mass. Body mass is typically taken as the predictor. Were these reversed regressions used to calculate the residuals or is this simply a mistake here? If done as shown, body mass would be corrected for clutch size rather than the other way around. Also, since residuals were calculated at the taxon level, it would help to either show the taxon-specific relationships or at least use different colors to better reveal the strength or relationships and the reliability of residuals (particularly for clutch size in the left half of the figure).

REVIEWER COMMENTS

Reviewer #1 (Remarks to the Author):

In this manuscript, Koçillari and colleagues tackle a big question using an innovative (to the field) approach applied across tetrapods. Specifically, the authors set out to explain the tremendous variation observed in sperm morphology and how this variation is shaped by body size. In addition, they assess other factors which are argued to influence sperm size evolution, including sperm competition risk, clutch size and genome size. The use of a Pareto Optimality approach to examine the evolution of reproductive traits is a novel addition to the field and something that I had to invest some time to read up on. The Pareto Optimality approach is particularly well suited for uncovering complex trade-offs among competing traits that constrain how traits move in trait morphospace. While the applicability of the Pareto Optimality approach to address biological questions has been highlighted for a little over a decade (at least if we take the Shoal et al. 2012 Science paper as the starting point), I'm not aware of this approach being applied to sperm evolution. The authors have done a great job in identifying the potential of this approach and applying it to sperm trait evolution, where trade-offs are expected to influence the evolution of trait values.

Using this approach, the authors demonstrate that the morphospace when assessing sperm length and body size is a triangle, which is indicative of a trade-off between multiple tasks with three points of optimality (archetypes). The authors then explore if this pattern is observed when contrasting species with different fertilization modes, thermoregulatory methods, and when assessing different tetrapod classes. Together, this is an interesting set of analyses and helps to make sense of the seemingly unclear pattern that emerges when plotting sperm length and body mass.

Response: *we thank the reviewer for the positive comments on our manuscript and on the Pareto Task Inference approach that we are applying here to evolutionary biology.*

Having suggested using Pareto plots that sperm length is likely subject to trade-offs is interesting but not entirely surprising – this is a common view but providing a robust demonstration is good to see. However, to further advance the field one would need to identify the traits driving these potential trade-offs. Here is where the manuscript fell short for me.

Response: *we thank the reviewer for the comment. We understand the point, but we partially disagree with the reviewer. There are two important considerations. Firstly, we provide here evidence, for the first time, and at an unprecedented, wide phylogenetic level (tetrapods), that long sperm length only evolves in small-to-intermediate-sized species, and that this pattern holds for the entire group, as well as for most classes analyzed separately (at least for those classes for which we have sufficient data). We therefore disclosed a macroevolutionary pattern that was never evidenced before. Tetrapods form a robust triangular Pareto front in the trait space of sperm length and body mass, indicating that the factors promoting the evolution of long sperm do not vary randomly within the observed variation in body mass across and within tetrapods. Again, this pattern has never been described at such wide taxonomic level. This is an important result and will promote further investigations to understand the association/constraint between factors influencing sperm evolution and body mass. Secondly, among the factors we considered, it was certainly not surprising that sperm length and the level of sperm*

competition, are positively associated. The influence of sperm competition on sperm length evolution has been shown in several studies at a lower taxonomic scale. Even if we dismiss the fact that the present study is the first to provide support for the sperm competition hypothesis at the level of tetrapods, we think that this result is important as a proof of the reliability of our methodological approach. For the other factors that we considered, we provide evidence of the role of clutch size in promoting the evolution of long sperm (see our response below). Moreover, we found that genome size is negatively correlated with sperm length. The association between genome size and sperm length has been rarely studied, and the predicted positive association between genome size and sperm length was not supported (e.g. Gage 1998). This hypothesis was based on the idea that genome size influences sperm head size, which in turn requires longer flagellum to attain sperm mobility (Gomendio and Roldan 1991). In contrast, we found that species with long sperm have smaller genomes. Considering that long sperm are associated with high levels of sperm competition and hence with gametic production, one interpretation may be that the large genomes may be more strongly selected against in species with high sperm production because of the longer meiotic duration associated with genome size (Bennett 1971). In the discussion we have largely expanded the interpretation of our results and we explained why these results are novel (lines 339-349).

The treatment of the literature and motivation for the factors being tested is hard to follow and incorrect at times (lines 70-86 in particular and throughout the manuscript). The manuscript identifies several factors that may influence the body size-sperm size relation, most of which I had an issue with. These include:

1. Mass-specific metabolic rate. This point has been made elsewhere, but it is challenging to disentangle MSMR with dilution effects since these two factors, both of which have been hypothesized to explain sperm length variation, covary with one another. However, after its introduction, MSMR was not included in the analyses, which was odd.

Response: *we agree with the reviewer in the fact that we were not clear enough in elucidating our hypothesis on MSMR. Since, as the reviewer stated, body mass and metabolic rate covary, we did not include MSMR in our analysis. We have now changed the introduction regarding this part to make our hypotheses clearer (lines 85-103).*

2. Clutch/litter size: The argument that clutch/litter size is positively correlated with sperm size in internal fertilizers is not supported by the citations provided. Similarly, in external fertilizers, the claim that larger clutches favor sperm number at the cost of size is not supported by the citations (also, your Stockley citation here should probably be Stockley et al. 1996, not 1997 as the latter paper doesn't look at female traits at all). Importantly, Stockley et al. 1996 shows a positive correlation between egg number and sperm length, which is contrary to the claims used to build the argument here. Most of the cited literature doesn't even mention clutch/litter size, which I found confusing. Overall, I'm not convinced there is good support at all for a clear link between clutch/litter size and sperm length (there is no correlation in most studies that have looked for a link, e.g., Byrne et al. 2003 Proc B). Sperm number is a different story – here there is evidence that larger clutches are correlated with larger testes but that is not the argument being advanced. Overall, this hypothesis lacked a clear connection to the literature and was poorly motivated, which left me wondering why it was included.

Response: we thank the reviewer for spotting the mistake and to provide feedback on our hypothesis. We have now changed the reference accordingly. We agree with the reviewer that evidence on the relationships between sperm length and clutch size are limited and mainly restricted to frog and fish species. However, traits associated with female reproductive biology are known to affect the evolution of sperm length in males (we provide now in the main text some references to support this). Our analyses span over four different classes, with both internal and external fertilization and where eggs can be fertilized simultaneously (externally in amphibians and internally in mammals) or sequentially (in birds and reptiles). We thus expect that the association between clutch size and sperm length may be due to different specific mechanisms. For example, clutch size may increase egg mass volume and hence selection for longer, more mobile sperm in the case of simultaneous fertilization; clutch size may instead increase the duration of female sperm storage and the opportunity for sperm competition in sequential fertilizers (e.g. birds). Even in external fertilizers, large clutches may require more time to be shed, increasing the opportunity for several males to fertilize the same batch of eggs. Although the selective pressure for longer sperm may therefore be due to different mechanisms, all these mechanisms converge to predict that clutch size and sperm length should covary, even across such a diverse group of species. We have revised the paragraph and we explained why a relationship between clutch size and sperm length can be expected also in other taxa, such as birds (lines 69-84).

Regarding the second part of the comment on sperm number, we agree that the association between sperm number and clutch size has been proposed and tested multiple times. Our results, however, demonstrate that long sperm have evolved in species with both large clutch size and large testes mass relative to body mass, evidence that was lacking before. We think that the large phenotypic variation considered in the present study has allowed to unveil previously unknown trade-offs. However, we acknowledge that other factors that we did not consider may contribute explaining interspecific differences in sperm length at a lower taxonomic scale.

3. Genome size: I regard this as a ‘zombie hypothesis’, one that one die despite a general lack of evidence! It seems compelling to argue that there should be a link between genome size and sperm length, and there is plenty of genome data to bring to bear on a large comparative analysis like this. But is there any evidence that sperm size is related with genome size? In mammals, some early work examining sperm evolution in a broad comparative context by Gage (1998, Proc B, 265:97-103) found no association between sperm head length or head area and chromosome number or genome mass. Similarly, in ferns, spore size is not related with genome size (Dyer et al. 2013, BMC Plant Biol, 13:219). Instead, genome size appears to be more often related with egg size (e.g. in fishes, Hardie and Hebert 2004, Can J Fish Aquat Sci, 61:1636-1646; in snapping shrimp, Hultgren et al. 2021, J Evol Biol, 34:1827-1839).

Response: we agree with the reviewer that we did not explain our argument well enough. We have now rewritten the paragraph (lines 85-103). The scientific reasoning to include genome size in our analysis came from the observed association between cell size and genome size. The association has been found in many organisms from yeast lineage to various multicellular organisms and we have added new references about that in the introduction. The point of the reviewer regarding the egg size is important. The fact that an association with egg size and genome size exists, justifies even more the

need to investigate the association between genome size and male gamete size. Accordingly, we found and cited in the text a recent review showing that the potential relationship between genome size and cell size extends also to gametes, including sperm (Glazier 2021). The lack of association between sperm size (either head size or total length) in previous work on a smaller taxonomic level may depend on the limited range of phenotypic variation considered in previous studies. For example, tetrapods show a larger range of variation in genome size and sperm length than the variation observed, for example, among mammals or birds only. Altogether our results suggest that long sperm do not evolve in species with very large genomes, suggesting that selection against very large genomes may be stronger in these species. This relationship may become apparent only when “extreme” phenotypes, in terms of both sperm length and genome size are considered and may explain why previous studies did not support the hypothesis.

Moreover, since genome size has been recently investigated in relation to metabolism rate, which ultimately affected cell size and cell activity, we have also explained the potential link between genome size, cell size and metabolism rate. Specifically, genomic size is indirectly linked to metabolic rate via its influence on cell size. Sperm cells with a bigger genome require a longer flagellum to compensate for the drag caused by the bigger sperm head, especially in large species. Sperm length in relation to body size may thus result from contrasting effects between metabolic rate and genome size. In large animals, low metabolic rate may constraint the evolution of sperm length while the large genome size may favour the evolution of long sperm. How these two contrasting hypotheses have shaped the evolution of sperm length in relation to body size in animals is unknown.

4. Sperm competition: I agree with this argument.

My take away from these hypotheses was that they lacked a clear motivation for why they would drive trade-offs in sperm length with body size. Without the authors can expand on their motivation it remains unconvincing to me. An alternative interpretation to their results are that larger species have larger genomes and that is why genome size is closer to T3, and testes mass is positively correlated with clutch size and sperm size and that is why these traits are maximised at T2. Under this alternative interpretation it isn't clear what is driving the trade-offs but instead you are detecting multicollinearity effects among predictors variables. If this thinking is wrong headed (which is entirely possible as I'm new to this form of analysis) then more explanation is warranted.

Response: *thanks for the comment. As we have stated in the introduction, several hypotheses have been proposed to explain the evolution of long sperm, each of which, is expected to be directly or indirectly associated with body size. We have now tried to better explain this in the revised version of the MS. The interpretation of the results, in particular causation, is another matter and we agree that different interpretations of the results may be provided. For example, due to the covariance between long sperm and larger clutches and testes, we cannot disentangle with the present analysis which of the two factors may have driven the evolution of long sperm: larger clutches may have driven the evolution of long sperm and increased the opportunity of sperm competition (and hence testes mass), or larger clutches may have increased the opportunity for sperm competition (and hence testes mass), which, in turn, may have favored the evolution of longer sperm. A similar consideration can be done for genome size, which appears to be constrained at T2 (long sperm) as compared to both T1 (small*

body size) and T3 (large body size). Across tetrapods, body size is therefore a poor predictor of genome size, which instead is negatively associated with sperm length, but also, and we agree with the reviewer, with clutch size and testes mass. For these reasons, the reviewer is right in saying that our analyses do not allow to establish the causal relationships underlying the coevolution of the morphological traits we investigated (discussed at lines 375-384). However, our key point is that our analyses provided evidence that the distribution of sperm length-body mass in tetrapods lied on a geometric shape (a triangle), suggesting that functional trade-offs underly these traits. The same geometric regularity has been found within classes (i.e., mammals and birds), indicating that, despite the large ecological and functional differences between classes, the same general evolutionary pattern was maintained. The importance and the novelty of our study, therefore, does not lie in the covariation of sperm length with testes mass and clutch size, but in the evidence that 1) long sperm only evolved in small-to-intermediate tetrapods, 2) species that, in this range of body size, evolved long sperm were characterized not only by having larger testes and clutches, but also 3) smaller genomes. While some causal links between traits can probably be excluded (we are not aware of any hypothesis predicting that a small genome may cause the evolution of long sperm, large testes and large clutches, and even less so intermediate body masses), other could be further explored. For example, small genomes may be favored in species with high sperm production (larger testes), which in turn evolve longer sperm; alternatively, selection for a smaller genome and hence a smaller sperm head may be stronger in long sperm that may be under stronger selection for sperm swimming performance.

We stated this point in the discussion (lines 377-384): “although testes mass and clutch size are both maximized in species with long sperm and intermediate small-to-intermediate body mass, our analysis does not allow us to disentangle whether large clutches have favoured the evolution of polyandry and longer sperm evolve in response to the associated increased level of sperm competition, or if instead long sperm have a fertilization advantage when clutches are large, and large clutches are independently associated with polyandry. A similar reasoning could be applied to the negative correlation between genome size and sperm length. Genome size could be constrained by the evolution of long sperm or by the evolution of large clutches, which covary with sperm length”. It is important to point out that although a triangular distribution is expected under a trade-off scenario, the Pareto framework cannot solve the cause-effect relation among different factors. Specifically designed studies are needed to investigate how the trade-offs emerged.

Finally, my understanding is that the Pareto Optimality approach emerged from engineering and economics – two fields that usually don’t take into account confounding effects of evolutionary history. Therefore, it wasn’t clear to me how much the shared evolutionary history of the species/clades examined will influence the results. The authors do attempt to control for phylogenetic bias with a randomization test. However, this is a fairly weak approach to correct for phylogenetic effect given that it was performed at the class and order level and that it was performed in the absence of branch length information. Did you consider using phylogenetic independent contrasts for each trait value and using these in the analyses? Without a more robust account for the evolutionary history of the species examined it is challenging to interpret the findings. I recognize that in Shoval et al.’s 2012 paper two of their examples are comparative (Darwin’s finches and bats – the latter of which is shown in your Figure 1). However, the lack of formal analyses of these data in Shoval et al.

foreshadows the challenges of dealing with phylogenetic data using the Pareto Optimality approach that I'm not convinced has been addressed yet.

Response: *we thank the reviewer for highlighting the importance of robustly addressing potential biases due to shared evolutionary history among tetrapods in assessing the robustness of the triangular Pareto front we identified. In the earlier version of our manuscript, we employed the SibSwap method to test the significance of the triangular Pareto front. The SibSwap method aims to control for phylogenetic dependencies in Pareto analyses (Adler et al. 2021). Briefly, the approach involves randomizing traits independently across species within phylogenetically related groups. In our study, we randomized the values of sperm lengths among closely related sibling species, and similarly for body masses. Previously, we classified species as siblings if they were in the same class or order. We then shuffled the traits of sperm length and body mass across sibling species for each class/order and we statistically compared the SibSwap-shuffled distributions with the actual distribution of tetrapods. Our results suggested that the triangular Pareto front was robust against phylogenetic dependence at both the class and order levels. However, our initial analyses did not incorporate branch length information. In agreement with the reviewer's comment, we have now included the branch length information from the phylogenetic tree.*

In order to assess the robustness of the Pareto front to phylogenetic dependencies among species, we have implemented several analyses in the revised version of the manuscript which resulted in a new figure (Figure 5). 1) We built the phylogenetic tree of tetrapods following the approach outlined by Kahrl and colleagues (Kahrl et al. 2021), which includes branch length information. 2) We evaluated the robustness and statistical significance of the triangular Pareto front by selecting multiple time points along the phylogenetic tree. Our findings strongly indicate that shared evolutionary history does not impact the triangular Pareto front up until approximately ~65 Mya, roughly coinciding with the Cretaceous-Paleogene extinction event around 66 Mya. We have incorporated the new results in the newly generated Figure 5 and revised the section discussing phylogenetic bias in the manuscript (paragraph "Robustness of the Pareto front: control for phylogenetic bias, lines 217-249). 3) We then decided to investigate more in detail the role of phylogenetic dependencies on the Pareto front from 65 Mya to the present day. We developed a modified version of the SibSwap approach by considering species within each sibling tip as independent data points in the trait space. The description of the modified version of the SibSwap approach and the associated results are reported in Supplementary Information and in the Supplementary Figure 5.

Finally, 4) we also investigated the impact of the sampling bias on the robustness of the Pareto front to phylogenetic dependencies. We assessed the statistical significance of the triangular distributions at varying dataset sizes by subsampling datapoints ranging from 5% to 80% of the whole dataset. We then tested for phylogenetic dependencies along the phylogenetic tree as described above. The triangular Pareto front is similarly robust to phylogenetic dependencies as with the whole dataset when including at least 30% of the datapoints. The new analyses and results are described in the main text (paragraph "Robustness of the Pareto front: finite-sampling bias", lines 251-272) and in the panel b of Figure 6. Overall, our results suggest that there is no strong phylogenetic bias affecting the significance of the triangular-shaped Pareto front.

Specific comments:

Line 160: instead of ‘population’ do you mean ‘species’ here?

Response: *Corrected.*

Reviewer #2 (Remarks to the Author):

This paper studies the tradeoffs in sperm evolution. To do so it applies Pareto task inference theory and uncovers evidence for three sperm tasks in tetrapods.

The paper is elegant and comprehensive.

Response: *we thank the reviewer for the positive feedback on our manuscript.*

I support publication of this paper, once the following suggestions for improvement are considered:

1. To shed further light on the tasks, please compare to previous work on three archetype life strategies in endotherms - shrew, bat and whale. To do so It would be important to show data when restricted only to endotherms. Then to compare to the mass -longevity triangle of Szekely 2016 . This paper provides additional life history traits that peak near each archetype. The bat archetype - protected niche including naked mole rat, primates etc, show . Small body size, large brain, small clutch size and large inter birth periods. It seems that one of the sperm archetypes is close to bats, and hence may be associated with the protected niche life strategy.

Response: *thanks for the comment. We are aware of the mass-longevity triangle of Szekely 2016. In the discussion we have now commented on the similarities between the previous work and our results, specifically regarding the clutch size (lines 329-334). Indeed, both the works indicate that small endotherms can both maximize and minimize clutch sizes, suggesting that clutch size does not linearly correlate with body mass, as previously thought. The finding highlights the power of Pareto Task Inference in detecting non-linear associations between traits. Regarding the protected niche life strategy, our T2 vertex is populated by different type of tetrapods (both frogs, birds and mammals) which are characterized by multiple different ecological adaptations. Among the species that are distributed very close to the vertex T2 we did not identify any specific ecological traits that can be associated to a protected niche life strategy.*

2. I am impressed by use of state of the art phylogenetic test for ParTI- better to use the original name “sibswap”. It would be good to add an additional test for mammals alone while flipping the closets species on the tree.

Response: *We now changed the name of the randomization approach when controlling for the phylogenetic dependencies to “SibSwap”, thanks for pointing it out. In the current version of the manuscript, we have now made an extensive analysis to assess the role of phylogenesis on the full dataset rather than on mammals alone (paragraph “Robustness of the Pareto front: control for phylogenetic bias, lines 217-249, and see also the new Figure 5). While an analysis dedicated to the full phylogenetic dependencies in mammals could be interesting per se, we believe that including an*

additional test exclusively for mammals would fall outside the scope of our paper, which focuses on tetrapods. We decided to put our efforts on 1) assessing the impact of the phylogeny on the full dataset (see the new Figure 5), 2) on developing a new analysis for evaluating the role of the phylogeny using subsampled datasets from the entire tetrapod dataset (paragraph “Robustness of the Pareto front: finite-sampling bias”, lines 251-272 and see also the new Figure 6b). Finally, we also proposed a modified version of the SibSwap approach to account for some limit cases where the application of the SibSwap method could be overly conservative (see the Supplementary Information and the new Supplementary Figure 5).

3. It would help with skeptics to be more careful in the intro- the triangle is not proof of tradeoffs, it is the trait space distribution expected if there are three tradeoffs.

4. Pareto optimality theory is usually based on plotting in a space whose axes are performances at known tasks. Here the authors plot data in trait space and infer the tasks. It's better to call this “Pareto task inference theory”. It's like an inverse problem on classical Pareto optimality theory.

Response: *thanks for the two above comments, we changed the text accordingly throughout the revised manuscript.*

Reviewer #3 (Remarks to the Author):

This study explored across tetrapods how body size variation could influence sperm length evolution via trade-offs between multiple variables such as sperm competition levels, clutch/litter size and genome size. To this end, the authors used Pareto optimality theory, which allowed them to consider multiple trade-offs simultaneously. They found the effects of body size on sperm length to be mediated primarily by sperm competition levels and clutch size.

Overall, this is a great way of dealing with broad and complex patterns of trait evolution, considering the oftentimes multivariate effects instead of singling out specific factors while ignoring others. The paper is timely and relevant in this regard. However, several important issues currently limit my enthusiasm for this study, as it is difficult to judge the correspondence between results and conclusions. These issues would need to be addressed before I can fully support the paper.

Response: *thanks for the positive feedback on our work. We have now revised the main text accordingly to the reviewer's comments. Below, we provide point-to-point responses for each comment.*

1. I find the current use of residuals very problematic for clutch size and testis mass (particularly the former). First, I am concerned about controlling for body size in these variables and then combining their residuals with absolute body size again in the same analyses. At least this is how I understand from the description that the analysis was done. If does not reflect the actual analysis, it may not be clear for other readers either, and the paper would benefit from a more detailed and careful description.

Second, even more importantly, I am concerned about combining the taxon-specific residuals in inter-taxonomic analyses. This approach essentially assumes that a value of 0.5 or -1.2 means the

same in amphibians, birds or any other taxon. However, residuals very much depend on the distribution of the data and where the regression line falls. A poor correlation would result in a wide spread of data points and an unreliable regression line (and hence residuals), whereas a tight association would generate in a narrow distribution of residuals.

So, unlike sperm length or body mass, each residual value depends on the distribution of the remaining species of the taxon, and so values are more difficult to compare between taxa. Further, regression lines can be sensitive to outliers, which then translates into shifts in residuals across the dataset. Even standardizing residuals may not necessarily resolve some of these issues.

At this point, were residuals calculated for each given dataset (e.g. where subsamples were taken) or were those of the original dataset used for all analyses, regardless of which species were included? This is important to know precisely because of the dependency of residuals on the given data distribution.

Response: *thanks for this comment, which we found very useful. Regarding clutch size, the reviewer is totally right. We have now done the enrichment analysis on the within-class z-scores, which does not correct for body mass variation. It is reassuring that results remained substantially unchanged (see Fig. 2b). Instead for testes mass, we maintained our initial approach of using residuals of testes mass on body mass. This is because there is ample evidence that residual testes mass on body mass closely reflects the level of sperm competition in vertebrates (see Lupold et al 2020). Citing Lupold et al 2020: “This prediction [i.e. that testes size reflects sperm competition level] is so widespread that testes size (correcting for body size) is commonly used as a proxy of sperm competition, even in the absence of any other information about a species’ reproductive behaviour”. We used class-specific regressions to calculate residuals in amphibians, reptiles, birds and mammals. Specifically, we calculated the class-specific residuals (RTS) based on the body mass and sperm length dataset, using available testes mass values (197 amphibian, 324 bird, 408 mammal, and 29 reptile species). We provided detailed information in the Supplementary Information about the regression analyses on testes mass. Q-Q plots of the residuals matched nearly perfectly the expected normal distribution (see Supplementary Fig. 4). However, to further explore the potential role of the outliers on the residuals, we re-calculated the log-log regressions after excluding some outliers from each class and compared the residuals without outliers (RTS_{outliers}) with the original RTS values. In all the classes, the correlation between RTS and RTS_{outliers} was > 0.999 (Supplementary Table 2). Since our estimates were not influenced by outliers, we left our analysis on the relative testes mass unchanged.*

2. The authors mention that they shuffled closely related species along the phylogeny to separate phylogenetic dependency. However, throughout the paper I found no mention of what phylogenetic trees were used to map the species on in the first place, nor how phylogenetic associations were accounted for in their ‘original’ analysis that the randomizations were compared to. If it is true that no formal phylogenetic control was undertaken, this would be very problematic and, quite frankly, inadequate. If phylogenetic covariance cannot be incorporated into the analytical approaches taken (e.g. as a correlation structure), could the authors use the phylogenetically independent contrasts instead, or some other form of phylogenetic correction?

Response: *we thank the reviewer for the stimulating comment. In our initial phylogenetic control, we employed the SibSwap method to control for phylogeny at the class and order level, without*

considering the branch length information. After grouping the species either by class or order, we shuffled the traits of sperm length and body mass across species within class/order and statistically compared the SibSwap-shuffled distributions with the distribution obtained with the original dataset. Our results suggested that the triangular Pareto front was robust against phylogenetic dependencies at both the class and order levels. However, we acknowledge that our initial method had some limitations. In agreement with the reviewer(s)' comments, we recognize the importance of taking into account the branch length information from the phylogenetic tree. As stated also above (response to reviewer #1), to assess the robustness of the Pareto front to phylogenetic dependencies among species, we have implemented several analyses in the revised version of the manuscript (paragraph "Robustness of the Pareto front: control for phylogenetic bias, lines 217-249), which are mainly depicted in the new Figure 5. 1) We built the phylogenetic tree of tetrapods following the approach outlined by Kahrl and colleagues (Kahrl et al. 2021), which includes branch length information (see Methods). 2) We evaluated the robustness and statistical significance of the triangular Pareto front by selecting multiple time points along the phylogenetic tree. Our findings strongly indicate that shared evolutionary history does not impact the triangular Pareto front up until approximately ~65 Mya, roughly coinciding with the Cretaceous-Paleogene extinction event around 66 Mya. We have incorporated the new results in the new Figure 5 and revised the section discussing phylogenetic bias in the manuscript. 3) We then decided to investigate more in detail the role of phylogenetic dependencies on the Pareto front from 65 Mya to the present day. We developed a modified version of the SibSwap approach by considering species within each sibling tip as independent data points in the trait space. The description of the modified version of the SibSwap approach and the associated results are reported in Supplementary Information and in the Supplementary Figure 5. Finally, 4) we also investigated the impact of the sampling bias on the robustness of the Pareto front to phylogenetic dependencies. We assessed the statistical significance of the triangular distributions at varying dataset sizes by subsampling datapoints ranging from 5% to 80% of the whole dataset. We then tested for phylogenetic dependencies along the phylogenetic tree as described above. The triangular Pareto front is similarly robust to phylogenetic dependencies as with the whole dataset when including at least 30% of the datapoints. The new analyses and results are described in the main text (paragraph "Robustness of the Pareto front: finite-sampling bias", lines 251-272) and in the panel b of Figure 6. Overall, our results suggest that there is no strong phylogenetic bias affecting the significance of the triangular Pareto front.

3. I often found the introduction and discussion somewhat confusing as the terminology, theory or patterns were freely moved between taxa to back up arguments, even though the authors themselves acknowledged that their biology can differ in important ways. Please double-check carefully where such argumentation is appropriate and where it might be too much of a stretch. In general, the discussion as a whole could be more streamlined and more on-point. My specific comments below point out some of these cases.

Response: we thank the reviewer for the feedback. Every taxon is indeed different with its own peculiar characteristics, which mainly reflect the great variation in life-history strategies. By analyzing this complexity overall, we aimed at understating whether a general pattern on the evolution of sperm length in relation to body mass arises despite the diversity of the taxa considered. We have now partly rewritten the introduction by also incorporating the feedback received by all the reviewers. Regarding

the discussion, we have largely expanded the interpretation of our results and we have tried to be more on-point with our explanations. Please see also the responses that we provide below.

Specific comments:

L24 “these factors” is unclear as it is too far removed from what it refers to. Consider reordering your statements.

Response: *corrected.*

L31 Since ‘triangular’ already means ‘shaped like a triangle’, so ‘triangular-shaped’ seems redundant (throughout the paper)

Response: *we have now limited our terminology to “triangular”.*

L53 shortest instead of smallest? Small and long use different dimensions and so are not direct contrasts.

Response: *the reviewer is correct, and we have now replaced it with “shortest”.*

L61 It is RELATIVE testis mass what is commonly used. This is important given the variation in body mass between species.

Response: *the reviewer is correct, we changed it accordingly throughout the manuscript.*

L66-67 In the same context, it might be important to consider the risk of sperm dilution as a constraint on sperm length that is less present in small species.

Response: *the reviewer is correct, and we have now rephrased the sentence accordingly.*

L80 though it seems to depend on the taxon according to Liao et al. 2018 (your ref no. 51)

L99 remove redundant “been”

Response: *corrected.*

L210 I am not convinced that the claim “our results are not affected by the phylogenetic dependencies” is correct. The phylogenetic effect may simply not have been strong enough to overpower the Pareto front (i.e. being weak), but it does not mean (in my opinion) that there was no phylogenetic dependency whatsoever. That said, I also do not understand where phylogenetic trees were even incorporated into the analyses.

Response: *in the revised manuscript, we were more specific in our statement and moved the previous analysis on the phylogenetic control at the class level to the new paragraph "Robustness of the Pareto Front: Finite-Sampling Bias" (lines 251-272). To thoroughly account for phylogenetic bias, we built*

a phylogenetic tree and, as described in our response to the second comment from the reviewer and also above, we employed the SibSwap approach by taking into account the branch length information. Our analysis demonstrated that the triangular Pareto front remained robust to phylogenetic dependencies until approximately 65 Mya, at which point the p-value exceeded the 0.05 threshold. However, we believe that the loss of significance is not related to phylogenetic bias but rather to the limitations of the SibSwap randomization approach. This approach may become overly conservative as we approach the terminal nodes because each sibling tip contains fewer and fewer species closer to the present day. Please refer to the section “Robustness of the Pareto Front: Control for Phylogenetic Bias Between 65 Mya and the Present Day” in the Supplementary Information and see Supplementary Figure 5. In summary, our results suggest that regardless of whether we consider older phylogenetic relationships among species (from 355 Mya to 65 Mya) or more recent ones (below 65 Mya), there is no strong phylogenetic bias affecting the significance of the triangular Pareto front.

L220 The Methods section is in an odd place. To my knowledge, this usually comes after the Discussion in Nature journals. Where it is now, it interrupts the flow of the paper.

Response: *we have now moved the methods to meet the format requirements.*

L222 I would consistently call this ‘sperm length’ (which it actually is), as sperm size sounds more like a 3-dimensional trait.

Response: *we agree with the reviewer, and we changed the terminology to “sperm length” throughout the MS.*

L251 What do you mean by “phylogenetically close?” Did you shuffle species within genera to compare the shuffled with the original data within a class? Or did you shuffle species between orders to do so or just randomly between all species of a class? This is not clear at all.

That said, please also note that the dataset contains crocodiles that would be more closely related to birds than to lizards and snakes... Given these are only three species, I am not worried that the outcome would change much, but it puts ‘closely related’ into perspective and how vague it is.

Finally, I see no mention of the phylogenies that were used and how the ‘original dataset’ was controlled for phylogeny, or how exactly species were shuffled across the phylogeny. If there was no phylogeny in the first place, how exactly did this shuffling address the phylogenetic dependency?

Response: *in the revised version of the manuscript, we added in the Methods a new section related to the building of the phylogenetic tree (in which indeed crocodiles are more closely related to birds than lizards/snakes) and another one with details of the SibSwap-shuffled approach. The new analyses and results are depicted in the new Figure 5. Please also refer to our responses above. Thanks again to the reviewer for the stimulating comments on phylogeny.*

L253-254 I am not clear about this definition of the p-value. If p represents the proportion of t ratios that are lower than the original (i.e. farther away from capturing

Response: *the reviewer is correct, and we have now revised and expanded the Methods section by providing more details related to the t-ratio (lines 427-444), which we also report below.*

“The t-ratio test to assess the statistical significance of Pareto fronts

A common approach used in Pareto analyses to assess the statistical significance of fitting Pareto fronts to data points is to compute the p-values using the t-ratio test (Hart et al., 2015; Hausser et al., 2019; Cona et al., 2019). The t-ratio defines the fraction between the area of the best-fitted polytope, computed through the Sisal algorithm (Bioucas-Dias, 2009), and the area of the convex hull that encapsulates the data points, computed through the “convhulln” algorithm in Matlab. If the distribution of data points in the trait space is triangular, the convex hull will resemble a triangular-shaped distribution, and its area will almost entirely fill the space within the triangular polytope that can be fitted on the data points. This means that the areas of the fitted triangular polytope and the convex hull will be similar, and the t-ratio will be close to one. In our dataset, we obtained a t-ratio of 1.03 (see Supplementary Fig. 2a). Conversely, randomly distributed data points will have a cloud-like shape in the trait space. This cloud of randomized data points will still be fitted by a triangle, but this time the convex hull will hardly occupy the space within the best-fitting triangular polytope, and the regions in space close to the vertices of the triangle will remain mostly empty. In this case, the t-ratios will be consistently larger than one. For instance, by randomizing the data points of our dataset we found a t-ratio of ~1.25 (see Supplementary Fig. 2b). Next, the p-values are defined as the proportion of instances where the t-ratios of the randomized datasets are lower than those of the original dataset, divided by the total number of randomized datasets. We considered as statistically significant p-values those that scored under 5% of times ($p < 0.05$).”

L258 It would be helpful to provide at least a minimal description of the methods rather than only referring to a separate paper. Readers should not be asked to search for papers (possibly even behind a paywall) to get at least an inkling of what was done.

Response: *Following the reviewer’s comment, we have now expanded our Methods section “The triangularity test, density enrichment analysis and the position of the archetypes”. We split the paragraph in three sections: 1) “The t-ratio test to assess the statistical significance of Pareto fronts” (see the content of it in the previous comment), 2) “Randomized datasets in the trait space” (lines 446-453), which content is reported below, and 2) “Feature enrichment of the archetypes” (lines 455-467), which content is reported below.*

“Randomized datasets in the trait space

In cases when the data points were considered as independent, the trait randomization was done by independently shuffling the values of sperm length and body mass across all points in the distribution. However, when phylogenetic correlations among data points become relevant, we followed the SibSwap randomization approach as described in (Adler et al., 2021). This method proposes randomly permuting the values of each trait independently (i.e., sperm length and body mass) within each set of terminal nodes sharing a parental node. This approach preserves both the phylogenetic constraints and the marginal distributions of each trait (Adler et al., 2021).”

“Feature enrichment of the archetypes

According to the Pareto Task Inference, archetypal species near the vertices should exhibit maximum values in a set of enriched features, while species far from the vertices should have lower score values in those features. The scores of such features should monotonically decrease with the Euclidean distance from the vertices. The features that we considered for this analysis were the continuous traits of testes mass, clutch/litter size, genome size. To investigate which feature enriches at the vertices of the triangle, we computed the density profile of each feature in terms of mean values as a function of the Euclidean distance from the archetypes. To do so, we first sorted data points using their Euclidean coordinates in trait space from the given vertex. We then divided the distribution of species in the trait space into 8 equally populated bins. In each bin, we computed, for the given feature, the mean value of that feature across the species within the given bin. We then defined as enriched features those that passed the p-value test based on the two-sample t-test computed between the scores of the feature at the bin nearest the vertex and the scores of the rest of the distribution of points excluding the species of the nearest bin.”

L276-315 This is a very long paragraph that could easily be topically split into two or three paragraphs for easier reading.

Response: *we agree with the reviewer, and we split this long paragraph in three. Thanks for the suggestion.*

L282 “while it is”

L283 no hyphenation in “small or large body sizes”

L284 insert ‘the’ after ‘that’

L288 delete ‘into’ after ‘entering’

Response: *correct.*

L289 I am not aware of either of the two cited papers reporting an association between the size of female sperm-storage organs and clutch size. Rather, a relationship was found between clutch size and the duration of sperm storage (i.e., essentially the laying period), which makes sense if one egg is laid per day in most species.

L289-292 This is a confusing sentence. First, it moves the discussion from interspecific relationships to the species level; second, it uses mammals with single offspring to back up the argument about body size, longevity and clutch size in birds. This line of argument is confusing: The association between body size and longevity is more likely to hold between than within species, as is the r vs. K reproductive strategy. Further, single offspring in mammals do not inform about clutch size in birds. Why not use the many long-lived birds (e.g. seabirds) that produce only 1-2 eggs per season (or possibly even only every other year)? Also, what seems entirely neglected in this line of argumentation is that parental care plays an important role (so it is not necessarily about longevity per se). Please also note that none of these animals produce only a single offspring – they just do so per reproductive bout. Finally, mixing ‘clutch’ and ‘mammals’ as if mammals also had clutches (rather than e.g. litters) further adds to the confusing argument. Please revise.

Response: *the reviewer is correct, and we apologize for not being clear in the first version of the manuscript. We have now re-written the paragraph. The specific part on birds has been rewritten and moved to the introduction section (lines 77-81). We also removed the part on reproductive strategies, which was indeed misleading. We have now entirely focused on discussing our results.*

L292-294 Curiously, this argumentation seems to apply to fish but not amphibians based on the very paper cited here. Hence, the result in amphibians may have need another explanation?

Response: *the reviewer is totally correct. Our results suggest an overall positive association between clutch size and sperm length. Although we cannot test this finding specifically in frogs, as the triangle is marginally not significant in frogs, we can nevertheless assume that our overall finding also applies to frogs. We added a brief explanation to interpret these results (lines 320-325). Thanks for the suggestion.*

L295 Note that sperm competition itself is not high or low, but its level can be. Alternatively, just say “strong/intense sperm competition” (or similar).

Response: *thanks for spotting the mistake, we have corrected it.*

L307 It is not clear why this negative result is particularly interesting given there has not been much evidence for it in previous studies already? Also, it is not clear how the conclusion about genome size being constrained in species with high sperm competition levels is derived from sperm length and body mass variation (previous sentence)? Please be more specific about how these different traits are linked.

Response: *the reviewer is correct in saying that we could be clearer in explaining the interaction between different traits. However, we do not consider the result on genome size as a negative result because we found that genome size is significantly maximized in large species (T3) and minimized in species with long sperm (T2). The result thus indicates that genome size is both associated with sperm length and body mass. We have now rewritten the paragraph to make the interpretation clearer (lines 337-349).*

L344 This sentence is confusing. Maybe a comma is needed after clutch size to separate those parts of the sentence? Does this then express what you intended?

Response: *we have now broken down the sentence, thanks for the suggestion.*

Supplementary Fig. 3: This is an unusual representation of the relationship between body mass and clutch size/testis mass. Body mass is typically taken as the predictor. Were these reversed regressions used to calculate the residuals or is this simply a mistake here? If done as shown, body mass would be corrected for clutch size rather than the other way around. Also, since residuals were calculated at the taxon level, it would help to either show the taxon-specific relationships or at least use different colors to better reveal the strength or relationships and the reliability of residuals (particularly for clutch size in the left half of the figure).

***Response:** the reviewer is correct, thanks for spotting out the mistake. The new plots are now shown in Supplementary Figure 3. We represented the relationship between body mass and testes mass in each class separately by taking the body mass as the predictor. For clutch size, since we decided to not use the residuals, as also suggested by the reviewer, we removed the plot.*

- Adler, M., A. Tandler, J. Hausser, Y. Korem, P. Szekely, N. Bossel, Y. Hart, O. Karin, A. Mayo, and U. Alon. 2021. Controls for Phylogeny and Robust Analysis in Pareto Task Inference. *Molecular Biology and Evolution* 39.
- Bennett, M. D. 1971. The duration of meiosis. *Proceedings of the Royal Society of London. Series B. Biological Sciences* 178:277-299.
- Gage, M. J. G. 1998. Mammalian sperm morphometry. *Proceedings. Biological sciences / The Royal Society* 265:97-103.
- Glazier, D. S. 2021. Genome Size Covaries More Positively with Propagule Size than Adult Size: New Insights into an Old Problem. *Biology (Basel)* 10.
- Gomendio, M. and E. R. Roldan. 1991. Sperm competition influences sperm size in mammals. *Proceedings. Biological sciences / The Royal Society* 243:181-185.
- Kahrl, A. F., R. R. Snook, and J. L. Fitzpatrick. 2021. Fertilization mode drives sperm length evolution across the animal tree of life. *Nature ecology & evolution* 5:1153-1164.

REVIEWERS' COMMENTS

Reviewer #1 (Remarks to the Author):

The authors have done a great job in addressing my concerns from the previous round of review. The motivation for the work and the analyses have been strengthened and the manuscript is much improved as a result. The inclusion of additional phylogenetic sensitivity analyses are great. The author also provide a balanced discussion of their results, taking time to stress the limitations of the approach and correlative nature of the results. highlighting numerous avenues for future exploration, which will no doubt help guide the field.

I congratulate the authors on a well presented set of analyses that shed light on the complex nature of trade-offs among ejaculate traits in tetrapods.

Specific comments:

Line 318: consider 'For clutch size, our results...'

Reviewer #1 (Remarks on code availability):

The link to the code did not work.

Reviewer #3 (Remarks to the Author):

I compliment the authors on a thorough revision that greatly improved their paper. I have no further comments except for recommending a few very minor edits to consider during proofreading.

47 Maybe state the taxa with very short sperm (e.g. some Hymenoptera) so the sentence does not read as if fruit flies covered that entire range of sperm lengths (they only hold the upper record).

88 explaining -> explain

116 Trade-offs can then be inferred...

370 front -> fronts

473 calibrating -> calibrate

Reviewer #3 (Remarks on code availability):

The link did not work, so I was unable to access the code.

Reviewer #1 (Remarks to the Author):

The authors have done a great job in addressing my concerns from the previous round of review. The motivation for the work and the analyses have been strengthened and the manuscript is much improved as a result. The inclusion of additional phylogenetic sensitivity analyses are great. The author also provide a balanced discussion of their results, taking time to stress the limitations of the approach and correlative nature of the results. highlighting numerous avenues for future exploration, which will no doubt help guide the field.

I congratulate the authors on a well presented set of analyses that shed light on the complex nature of trade-offs among ejaculate traits in tetrapods.

Response: we are delighted to receive this positive feedback on the revised version of our MS. Thanks again for the useful suggestions provided during the first round.

Specific comments:

Line 318: consider 'For clutch size, our results...'

Response: changed as suggested.

Reviewer #1 (Remarks on code availability):

The link to the code did not work.

Response: we shared a private link, we are sorry for having overlooked this.

Reviewer #3 (Remarks to the Author):

I compliment the authors on a thorough revision that greatly improved their paper. I have no further comments except for recommending a few very minor edits to consider during proofreading.

Response: we are delighted to receive this positive feedback on the revised version of our MS. Thanks again for the useful suggestions provided during the first round.

47 Maybe state the taxa with very short sperm (e.g. some Hymenoptera) so the sentence does not read as if fruit flies covered that entire range of sperm lengths (they only hold the upper record).

Response: we changed accordingly the suggestion, thanks (line 45)

88 explaining -> explain

Response: corrected

116 Trade-offs can then be inferred...

Response: corrected

370 front -> fronts

Response: corrected

473 calibrating -> calibrate

Response: corrected

Reviewer #3 (Remarks on code availability):

The link did not work, so I was unable to access the code.

Response: we shared a private link, we are sorry for having overlooked this.